# Myosin-dependent cell-cell communication controls synchronicity of division in acute and chronic stages of *Toxoplasma gondii*

Karine Frénal[1], Damien Jacot[1], Pierre-Mehdi Hammoudi[1], Arnault Graindorge[1,†], Bohumil Maco[1] & Dominique Soldati-Favre[1]

The obligate intracellular parasite *Toxoplasma gondii* possesses a repertoire of 11 myosins. Three class XIV motors participate in motility, invasion and egress, whereas the class XXII myosin F is implicated in organelle positioning and inheritance of the apicoplast. Here we provide evidence that TgUNC acts as a chaperone dedicated to the folding, assembly and function of all *Toxoplasma* myosins. The conditional ablation of *TgUNC* recapitulates the phenome of the known myosins and uncovers two functions in parasite basal complex constriction and synchronized division within the parasitophorous vacuole. We identify myosin J and centrin 2 as essential for the constriction. We demonstrate the existence of an intravacuolar cell–cell communication ensuring synchronized division, a process dependent on myosin I. This connectivity contributes to the delayed death phenotype resulting from loss of the apicoplast. Cell–cell communication is lost in activated macrophages and during bradyzoite differentiation resulting in asynchronized, slow division in the cysts.

[1] Department of Microbiology and Molecular Medicine, CMU, University of Geneva, 1 Rue Michel-Servet, 1206 Geneva, Switzerland. † Present address: Laboratoire de Biologie Cellulaire et Moléculaire, Faculté des Sciences Pharmaceutiques et Biologiques, 15 Avenue Charles Flahault, 34093 Montpellier Cedex 5, France. Correspondence and requests for materials should be addressed to D.S.-F. (email: dominique.soldati-favre@unige.ch).

Toxoplasma gondii, the causative agent of toxoplasmosis, belongs to the phylum of Apicomplexa that includes relevant human pathogens such as *Plasmodium* species responsible for malaria. Members of this phylum are obligate intracellular parasites that actively invade and egress from their target host cells using a substrate-dependent locomotion called gliding motility. The glideosome refers to the actomyosin system that powers forward movement of these parasites. It is located between the inner membrane complex (IMC) and the plasma membrane and composed of the class XIVa MyoA and gliding-associated proteins (GAPs)[1]. MyoA is conserved throughout the phylum[2] and its depletion in *T. gondii* and *P. berghei* critically impacts motility, invasion and egress[3–6]. Conserved in the coccidian subgroup of Apicomplexa, the class XIVc TgMyoH initiates motility at the apical tip of the parasite likely by translocating adhesin-receptor complexes from the apex to the beginning of the IMC where TgMyoA takes the relay along the pellicle[7]. The class XIVb TgMyoC also restricted to the coccidians partially compensates for the deleterious but not lethal ablation of TgMyoA by its re-localization from the basal polar ring to the pellicle[4,8,9]. The class XXII TgMyoF is conserved across the phylum and indispensable for the correct positioning of the two centrosomes during cell division and the segregation of the apicoplast, a non-photosynthetic plastid-like organelle[10]. TgMyoF is additionally involved in the directed transport of secretory organelles called dense granules to the plasma membrane[11] and positioning of the rhoptries to the parasite tip[12]. In parasites depleted in TgMyoF, organelles accumulate in enlarged residual bodies (RB)[10], a structure derived from the mother cell postulated to contribute to the intravacuolar organization in rosette and as disposal of the remnants of the mother cell following division[13]. Parasites depleted in TgMyoF exhibit a 'delayed death phenotype' (DDP)[10]. This phenomenon described in *T. gondii* and *Plasmodium* spp., refers to perturbations of apicoplast functions by either pharmacological compounds or molecular genetic manipulation that result in loss of the organelle and in parasite death only on entry into the next lytic cycle[14–16]. *T. gondii* tachyzoite division is characterized by the synchronous formation and geometric expansion of two daughter cells within a mature mother cell; a process referred to endodyogeny[17]. Tachyzoites within a given vacuole tend to divide in a perfectly synchronized fashion, controlled by an unknown mechanism[18].

Among the Apicomplexa, *T. gondii* possesses the largest repertoire of genes coding for myosin heavy chains with 7 of the 11 motors being uncharacterized to date. TgUNC is a myosin-specific co-chaperone of the UCS (UNC-45/CRO1/She4p) family[19], which is critical for the heterologous production of soluble and functional TgMyoA[20]. UCS proteins interact with myosins of classes I, II, V and contribute to their stability[21] raising the question about the importance of TgUNC in the folding of the 11 *T. gondii* myosins belonging to five distinct classes[2]. Like in other eukaryotes except fungi, TgUNC presents three tetratrico-peptide repeats (TPR) at its N-terminus followed by armadillo repeats that constitute the central and the UCS domains[19,22]. The UCS domain binds the myosin head domain while the TPR domain is able to interact with the general chaperone HSP90 or HSP70 (refs 23,24). The function of TgUNC of producing soluble and functional TgMyoA in insect cells was not dependent on the presence of the TPR motifs[20] as similarly reported in *Caenorhabditis elegans*[25].

In the present study, we demonstrate that conditional depletion of TgUNC results in the destabilization of the 11 myosins and complete block of parasite propagation. In addition to the severe defect in motility, invasion, egress and alteration in apicoplast inheritance, TgUNC-depleted parasites fail to constrict their basal pole and divide asynchronously within the parasitophorous vacuole (PV). To identify the myosins responsible for these two uncovered functions, each uncharacterized motor was functionally dissected. TgMyoJ, located at the posterior pole, is involved in the constriction of the basal complex and its deletion in the cyst-forming type II strain leads to a complete loss of virulence in the mouse model of infection. TgMyoI, located in an intercellular network as part of the RB, participates in the intravacuolar synchronicity of parasite division. Critically, TgMyoI ensures cell–cell communication between intravacuolar tachyzoites, ensuring the synchronicity of division and shedding light on the mechanism of DDP. During stage conversion into bradyzoites and cyst formation, the connection between parasites is gradually lost and absent in cysts isolated from infected mice that replicate asynchronously.

## Results

**TgUNC ensures the assembly of all *T. gondii* myosins.** The co-chaperone TgUNC was first conditionally knocked down via U1 snRNP-mediated gene silencing with the concomitant epitope tagging of the gene at the endogenous locus[26] (Supplementary Fig. 1a). UNC-3Ty migrated at the expected size of 130 kDa (Fig. 1a and Supplementary Fig. 11), was cytosolic (Fig. 1b) and almost fully extractable in presence of PBS (Supplementary Fig. 1b). The transient transfection of a vector expressing the Cre recombinase[27] led to TgUNC depletion and loss of excised parasites after a few passages preventing the isolation of clones (Supplementary Fig. 1a). Next, CRISPR/Cas9 genome editing strategy was used to favour the replacement of the endogenous promoter with a tetracycline-repressive promoter (Supplementary Fig. 1c). The MycUNC-iKD clone was tightly downregulated with no TgUNC detectable by western blot (WB) analysis following 24 h of parasite cultivation in the presence of anhydrotetracycline (ATc) (Fig. 1c and Supplementary Fig. 11). ATc-treated parasites were severely affected in one or several steps of the lytic cycle since no lysis plaques were formed after 7 days (Fig. 1d). The defect was fully rescued by expressing either a second copy of TgUNC (UNC-Ty) or a truncated version lacking the 3 TPR motifs (UNCΔTPR-Ty), stably integrated under ATc selection (Supplementary Fig. 1d). Both UNC-Ty and UNCΔTPR-Ty localized to the cytosol (Supplementary Fig. 1e) and the inducible MycUNC was still regulated in complemented parasites (Supplementary Fig. 1f).

To determine their fate, all myosin heavy chains, except TgMyoA and TgMyoD, were C-terminally tagged at their endogenous locus in the MycUNC-iKD strain. TgMyoA was detected with specific α-TgMyoA antibodies, whereas for TgMyoD, antibodies detecting its specific myosin light chain TgMLC2 were used since TgMLC2 is known to readily disappear in the absence of TgMyoD[28]. On TgUNC depletion, all myosins were destabilized based on WB analysis, yet exhibiting different kinetics (Fig. 1e and Supplementary Fig. 11). In case of TgMyoD, a rapid disappearance of TgMLC2 was observed with almost undetectable level already 24 h after ATc treatment. TgMyoA, TgMyoE and TgMyoK were undetectable at 48 h after ATc treatment, while TgMyoC, TgMyoF, TgMyoG, TgMyoH, TgMyoI, TgMyoJ and TgMyoL levels were reduced but still detectable at this time point. Complementation of MycUNC-iKD with either UNC-Ty or UNCΔTPR-Ty restored the assembly of TgMyoA and presumably all the other myosins (Supplementary Fig. 1g).

Taken together, TgUNC is a cytosolic myosin-specific chaperone essential for the tachyzoite survival. The N-terminal TPR domain of TgUNC is dispensable for the ubiquitous role of TgUNC in the folding and stability of all the parasite myosins.

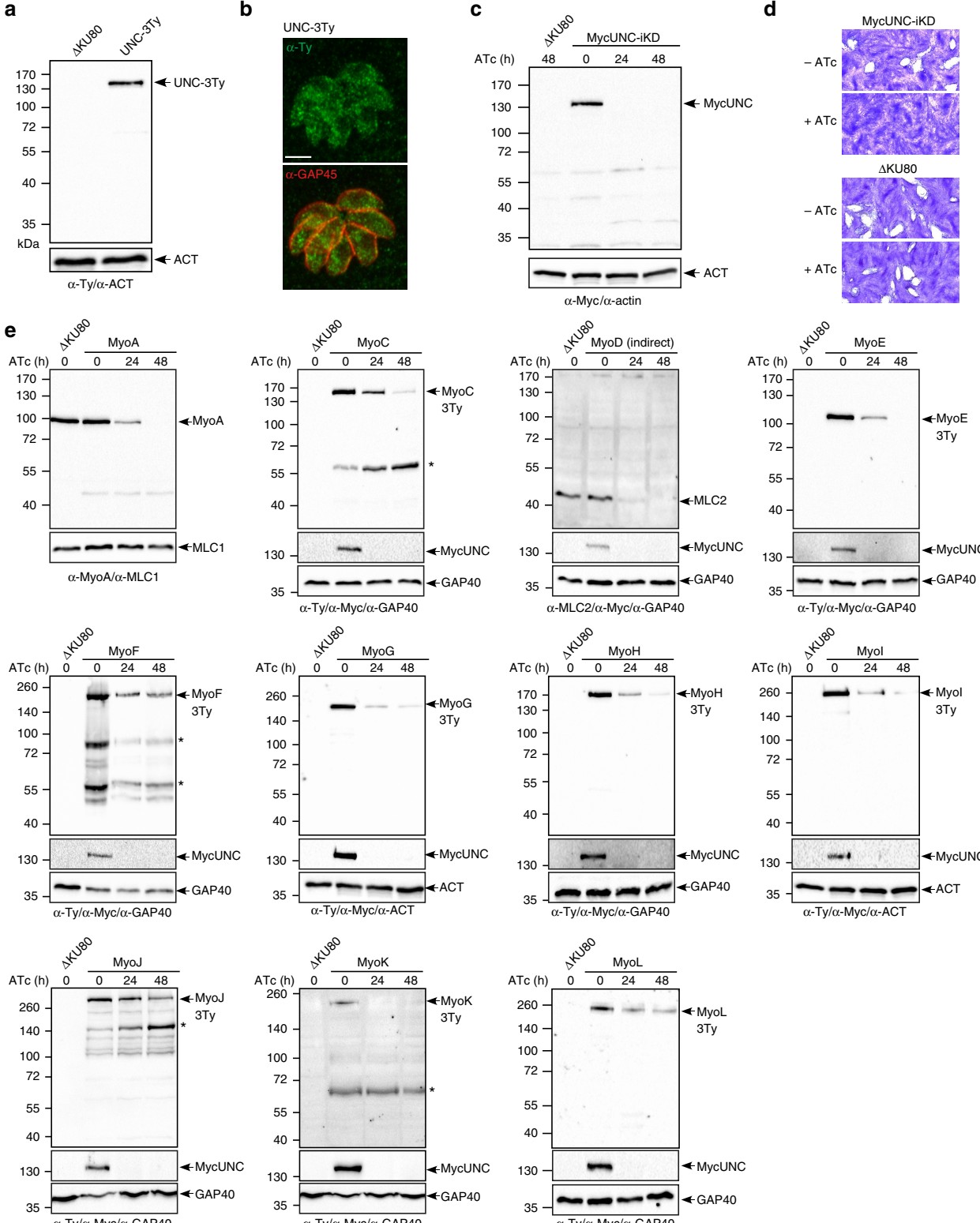

**Figure 1 | TgUNC is a myosin chaperone essential for tachyzoite survival. (a)** WB performed on total extract of extracellular tachyzoites, wild type (ΔKU80) or expressing the endogenously tagged TgUNC (UNC-3Ty). Actin (ACT) was used as loading control. **(b)** UNC-3Ty was detected in the cytosol of intracellular tachyzoites co-stained with the peripheral marker GAP45. Scale bar, 2 μm. **(c)** The regulation of the Tet-inducible cell line of TgUNC (MycUNC-iKD) was assessed by WB on total extract of extracellular tachyzoites. ACT was used as loading control. **(d)** TgUNC depletion has a severe impact on parasite survival as shown by plaque assay treated for 7 days ± ATc. **(e)** All classes of myosin heavy chains are destabilized on TgUNC depletion. All the myosin motors, except TgMyoA and TgMyoD, have been endogenously tagged with 3xTy in the MycUNC-iKD strain and their expression level was followed by WB on total extracts of extracellular tachyzoites treated or not with ATc for 24 or 48 h. Expression of TgMyoD was followed using α-MLC2 antibodies. GAP40 or ACT were used as loading controls. MycUNC-iKD was tightly regulated in all samples. Stars indicate myosin subproducts. Uncropped gels are presented in Supplementary Fig. 11.

**TgUNC depletion sums up the phenome of the known myosins.**
No defect in intracellular growth was observed 48 h after MycUNC depletion (Fig. 2a). In contrast and as anticipated from the rapid destabilization of TgMyoH, TgMyoA and TgMyoC, a severe block was observed in a gliding assay on poly-L-lysine-coated coverslips (Fig. 2b), as well as in invasion (Fig. 2c) and in egress (Fig. 2d) with less than 10% of entry and less than 5% of ruptured infected cells, respectively. Importantly, the PV membrane (PVM) was clearly ruptured attesting of the release of the perforin TgPLP1 and the lack of impact on microneme exocytosis (Supplementary Fig. 2a). In addition, TgMyoC, which is normally found exclusively at the basal polar ring, re-localized along the pellicle on TgUNC depletion (Supplementary Fig. 2b). This is in concordance with the differential kinetics of TgMyoA and TgMyoC destabilization (Fig. 1e) and the re-localization of TgMyoC while TgMyoA is deleted[29]. The myosin light chain TgMLC1, which is shared by TgMyoA, TgMyoC and TgMyoH, was only partially downregulated at 48 h (Fig. 1e).

The apical structure composed of spirally arranged fibres made of tubulin and termed conoid was previously reported to protrude in motile parasites in a $Ca^{2+}$- and actin-dependent manner[30,31]. Extracellular parasites depleted in MycUNC or treated with the actin depolymerizing agent cytochalasin D (CytD) still protruded their conoid on $Ca^{2+}$-ionophore treatment, arguing against the involvement of an actomyosin system in this process (Supplementary Fig. 2c).

TgUNC-depleted parasites showed no obvious division defect or alteration in biogenesis and positioning of micronemes, rhoptries, dense granules or mitochondrion (Supplementary Fig. 2d). However, these parasites failed to adopt the typical arrangement in rosettes and were clearly disorganized within their PV (Supplementary Fig. 2d), a phenomenon previously observed on TgMyoF depletion[10]. A defect in apicoplast inheritance was visible by indirect immunofluorescence assay (IFA) using antibodies against the luminal plastid chaperone TgCpn60 (ref. 32) (Fig. 2e) with 30% of the vacuoles containing parasites lacking the apicoplast (Fig. 2f). In a less pronounced manner compared to TgMyoF depletion, parasites depleted in TgUNC also divided asymmetrically with the two daughter cells growing in opposite direction (up and down) instead of growing in the same orientation towards the apical end of the mother cell (Supplementary Fig. 2e). The modest phenotype is likely explained by the inefficient destabilization of TgMyoF even 48 h after TgUNC depletion (Fig. 1e). Collectively, TgUNC depletion recapitulates the phenome of the known myosins.

**Basal constriction and synchronized division require TgUNC.**
Strikingly, the localization of TgMyoC in TgUNC-depleted parasites revealed an enlarged basal polar ring of mature parasites despite the absence of a cytokinesis defect (Fig. 2g). Using MyoC as marker, the diameter of the basal pole was measured and shown to increase by 1.8-fold in parasites treated with ATc for 48 h (Supplementary Fig. 2f). The earliest marker of the basal complex, TgMORN1, appears just after the duplication of the centrosomes as a ring-like structure capping the basal end of the developing daughter cells, which then contracts and ultimately caps the basal pole of the mature parasites[33,34]. TgMORN1 was endogenously GFP-tagged at its N-terminus in MycUNC-iKD parasites and shown as a dot at the centrocone, weakly at the apical end and strongly at the basal complex in non-dividing parasites (Fig. 2h). On TgUNC depletion, GFP-MORN1 localization was unchanged, however its basal staining delineated a larger ring that failed to contract. The EF-hand-containing protein centrin 2 (TgCEN2) is also known to be associated to the basal complex[34] and was C-terminally YFP tagged at the endogenous locus in MycUNC-iKD parasites

(Fig. 2i). TgCEN2 was visible at the apical end and annuli, and at the centrosome, however, the signal at the basal cup was absent in the absence of TgUNC. The morphological defect at the basal pole was examined by transmission electron microscopy (TEM). The basal complex appears as an electron-dense zone at the posterior ends of the IMC[35] and such a zone was visible in the absence of TgUNC; however, the gap between the basal ends of the IMC was considerably larger compared to wild-type parasites confirming the impairment in constriction (Fig. 2j).

A second prominent defect on TgUNC depletion was revealed by IFA performed with antibodies against TgISP1 and TgIMC1, two proteins localized to the apical cap and the IMC of mature and daughter parasites, respectively. Remarkably, the high synchronicity of parasite division inside a given vacuole that is typically observed in wild-type tachyzoites was lost in MycUNC-iKD treated 48 h with ATc (Fig. 2k). The α-ISP1 and α-IMC1 were used to score progression of daughter cell formation and a loss in synchronicity was reported in at least 30% of TgUNC-depleted parasites (Fig. 2l).

Taken together, TgUNC-depleted parasites uncovered two new myosin-associated processes, the basal complex constriction and the synchronized division of intravacuolar parasites.

**TgMyoI and TgMyoJ localize to the basal end of tachyzoites.**
The localization of uncharacterized myosins was determined by IFA following C-terminal insertion of 3Ty-epitope tags at the endogenous loci (Fig. 3a). TgMyoE was located to the conoid of daughter and mature parasites, whereas the unclassified TgMyoL was found to the conoid and cytosol. TgMyoG was weakly detectable at the periphery of the parasites. TgMyoK fluctuates along the cell cycle with a clear staining at the centrocones at the time when they duplicate as shown by its co-localization with TgMORN1. TgMyoI was found mainly in the RB, positioned predominantly at the centre of the rosettes. TgMyoJ was restrictively localized to the basal pole of developing daughters and mature parasites. The ring-like staining of TgMyoJ delineates the posterior end of the growing IMC that eventually constricts into a dot corresponding to the basal cup as shown by its co-localization with TgCEN2 (Fig. 3a).

Individual myosin knockouts were generated in the ΔKU80 parasites by interrupting the genes within the N-terminal head domain (Supplementary Fig. 3a,b). *TgMyoE, G, I, K* and *L* genes were individually disrupted without noticeable impact on parasite fitness, as monitored by plaque assay compared to their corresponding tagged version (Fig. 3b). Only MyoJ-KO parasites exhibited smaller plaques, indicative of a loss in fitness as confirmed by competition assay (Fig. 3c). Taken together, the presence of TgMyoJ at the basal pole and TgMyoI in an intercellular network suggests roles in basal constriction and parasites connection, respectively. Although it is not possible to formally exclude functional redundancies or adaptation mechanisms, the myosins characterized here are dispensable with only TgMyoJ necessary for optimal fitness.

**TgMyoJ is implicated in the basal complex constriction.** The positioning of the organelles in the myosin knockout mutants was not altered (Supplementary Fig. 4a); however, both MyoI-KO and MyoJ-KO parasites failed to develop into organized rosettes. In contrast to TgMyoF depletion, this disorganization did not correlate with an altered position of daughter cells growing in the up-and-down orientation during division (Supplementary Fig. 4b). IFA performed with α-IMC1 antibodies revealed a considerably enlarged posterior pole only in MyoJ-KO parasites (Fig. 4a and Supplementary Fig. 4a). Moreover, TgCEN2 was undetectable at the basal cup of MyoJ-KO and TgMyoC staining

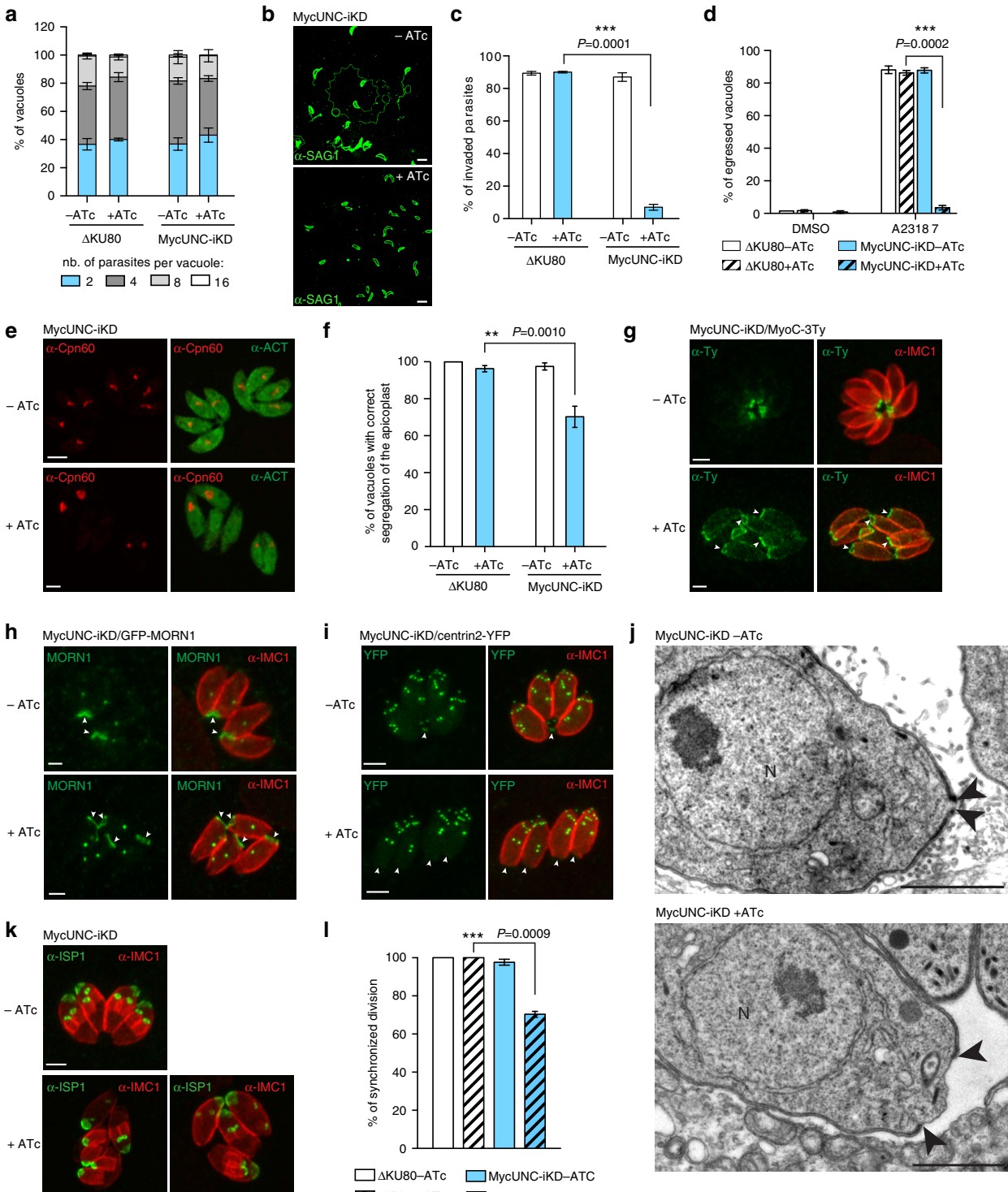

**Figure 2 | Impact of TgUNC depletion on tachyzoites.** (**a**) TgUNC depletion has no impact on intracellular growth. The number of parasites per vacuole was determined at 24 h after 48 h ± ATc. (**b**) TgUNC-depleted parasites are not able to glide as shown by the ionophore-induced gliding assay performed after 48 h ± ATc. Scale bar, 5 μm. (**c**) The invasion capacity of ΔKU80 and MycUNC-iKD strains was evaluated using a two-colour IFA performed after 48 h ± ATc. Parasites were allowed to invade for 30 min before fixing. (**d**) Ionophore-induced egress assay of ΔKU80 and MycUNC-iKD strains performed by treating the parasites with DMSO or A23187 for 7 min after 56 h ± ATc. (**e**) The presence of the apicoplast was assessed in MycUNC-iKD parasites using α-Cpn60 antibodies after 48 h ± ATc. Scale bar, 2 μm. (**f**) A slight defect in apicoplast inheritance was monitored in TgUNC-depleted parasites. (**g–i**) The posterior pole (arrowheads) of intracellular tachyzoites was examined using the endogenously tagged MyoC (**g**), MORN1 (**h**) or centrin2 (CEN2) (**i**) in the MycUNC-iKD strain treated ± ATc (48 h). Scale bar, 2 μm. (**j**) Electron micrographs of the basal pole of MycUNC-iKD parasites ± ATc (48 h). The arrowheads point to the basal ends of the IMC. N: nucleus. Scale bar, 1 μm. (**k–l**) The synchronicity of division was followed using α-ISP1 and α-IMC1 antibodies that stain the apical cap and the rest of the IMC, respectively (**k**) and quantified in ΔKU80 and MycUNC-iKD strains at 24 h after 48 h ± ATc (**l**). For **a,c,d,f** and **l**, the results are represented as mean ± s.d. from three independent experiments. Their significance was assessed using a parametric paired t-test and the two-tailed P-values are written on the graphs when significant.

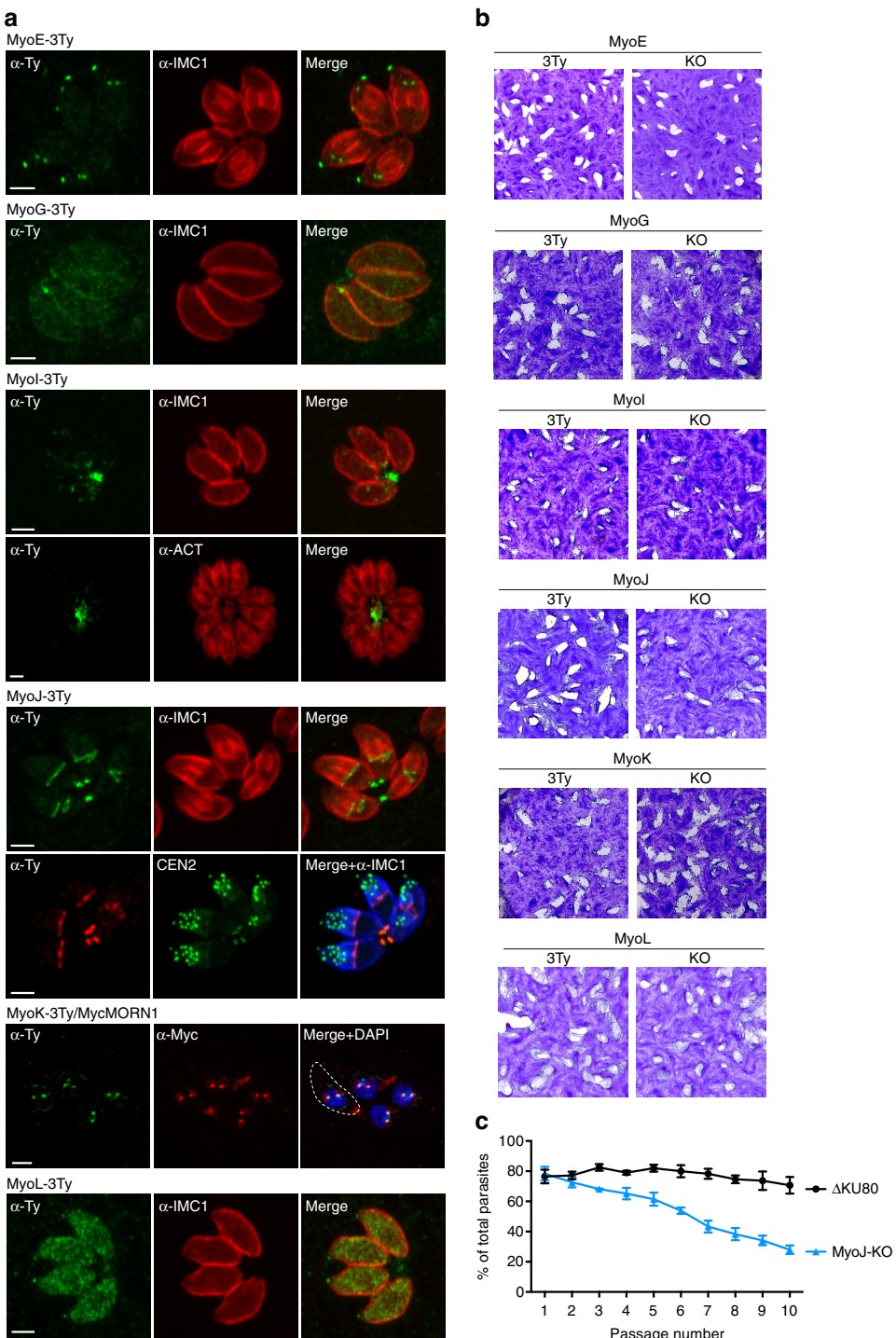

**Figure 3 | Localization and essentiality of myosin heavy chains in tachyzoites. (a)** Subcellular localization of the myosin heavy chains endogenously tagged with 3xTy. The peripheral marker α-IMC1 and the cytoplasmic marker α-ACT were also used to visualize the parasites. In addition, TgCEN2 and TgMORN1 were endogenously tagged with YFP and Myc, respectively, in the MyoJ-3Ty and the MyoK-3Ty background. Scale bar, 2 μm. Dashed lines represent the parasite periphery. **(b)** Plaque assays performed over 7 days on the 3Ty-tagged (3Ty) and knockout (KO) strains of myosins. No defect of fitness was monitored in the KO strains comparatively to the corresponding tagged strains except for MyoJ-KO as confirmed by competition assay using GFP-expressing parasites as an internal control **(c)**. The results are presented as mean ± s.d. from three independent experiments.

appeared as enlarged as observed in the absence of TgUNC (Fig. 4a and Supplementary Fig. 2f). TEM confirmed the defect in constriction in the absence of TgMyoJ (Fig. 4b). Furthermore, the F-actin-binding protein coronin (TgCOR), previously shown to accumulate to the posterior pole of extracellular parasites on $Ca^{2+}$-ionophore treatment[36] still re-localized at the basal pole in MyoJ-KO, indicating that TgMyoJ is not responsible for TgCOR re-localization in motile parasites (Supplementary Fig. 4c). TgMyoI was 3Ty-epitope tagged at the endogenous locus in MyoJ-KO and shown to accumulate at the enlarged posterior pole of the parasites but not detectable anymore in the RB (Fig. 4c). Simultaneous deletion of TgMyoJ and TgMyoI (MyoI/J-KO,

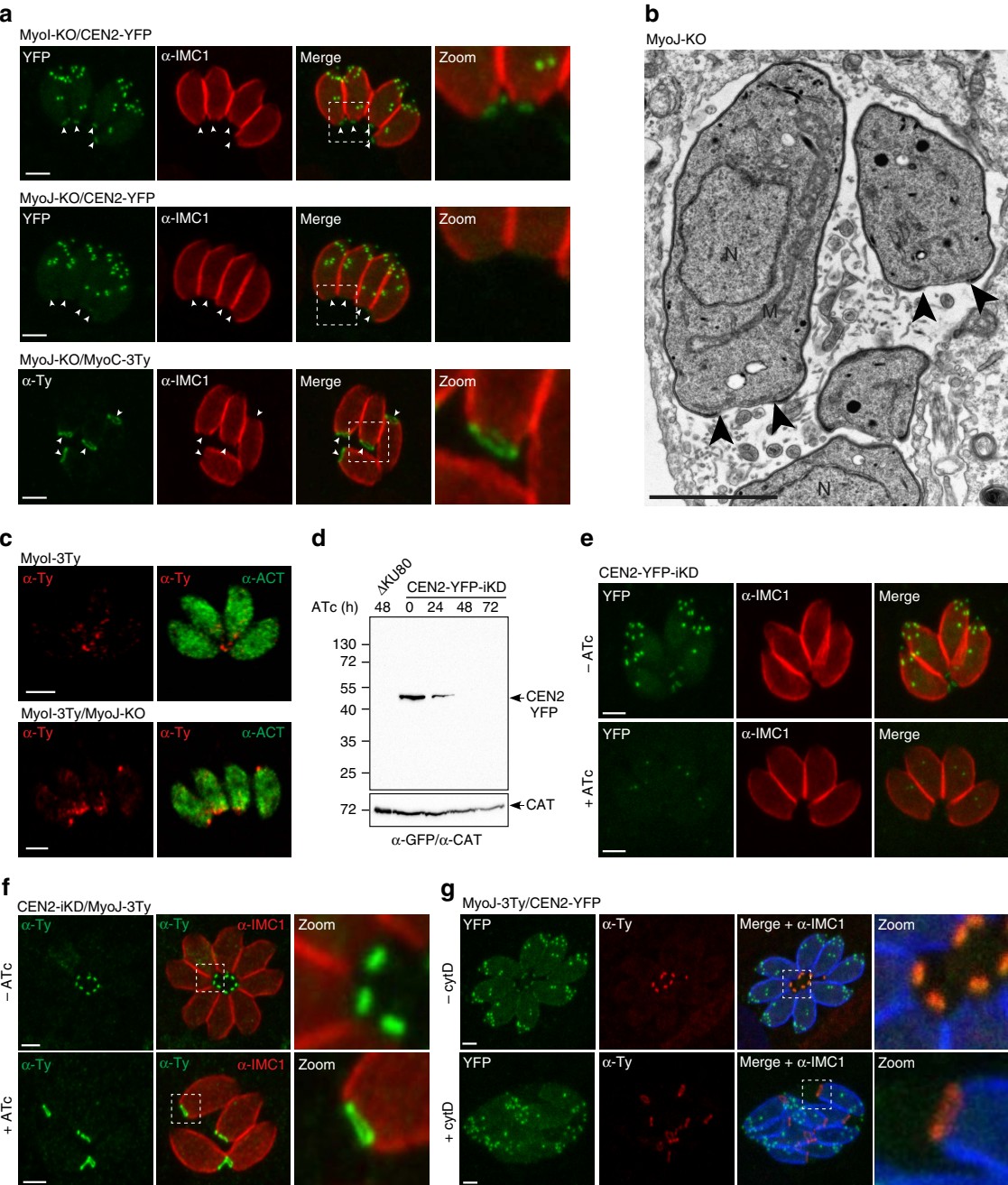

**Figure 4 | TgMyoJ and TgCEN2 are involved in the contraction of the basal complex.** (**a**) The signal of the endogenously tagged TgCEN2 (arrowheads) is not detected at the basal pole of MyoJ-KO in contrast to the one of MyoI-KO. In addition, the staining of endogenously tagged MyoC-3Ty (arrowheads) appears wider than in control parasites (Fig. 2g). Scale bar, 2 μm. (**b**) Electron micrographs of intracellular MyoJ-KO parasites. The arrowheads point to the basal ends of the IMC. N: nucleus, M: mitochondrion. Scale bar, 2 μm. (**c**) In MyoJ-KO parasites, the localization of TgMyoI is impaired and accumulates at the basal pole, inside the parasite, instead of being located at the residual body. Scale bar, 2 μm. (**d**) The regulation of the Tet-inducible TgCEN2 cell line (CEN2-YFP-iKD) was assessed by WB using α-GFP. Catalase (CAT) was used as loading control. Uncropped gels are presented in Supplementary Fig. 11. (**e**) The signal of CEN2-YFP-iKD was followed by IFA after 48 h ± ATc. Scale bar, 2 μm. (**f**) In absence of CEN2, the signal of MyoJ is detectable at the basal pole but delineates a wider ring. The diameter of MyoJ ring is 0.88 ± 0.03 μm (mean ± s.d.) in untreated CEN2-YFP-iKD ($n = 32$) and 1.48 ± 0.13 μm after 48 h under ATc ($n = 34$). The two-tailed $P$-value is < 0.0001 (****). Scale bar, 2 μm. (**g**) Intracellular MyoJ-3Ty/CEN2-YFP treated ± cytD (1 μm for 4 h). In treated parasites, MyoJ staining is wider (0.58 ± 0.05 μm, $n = 25$) than in untreated parasites (1.18 ± 0.12 μm, $n = 33$) and the staining of CEN2 at the posterior cup is not detectable. The two-tailed $P$-value is < 0.0001 (****). Scale bar, 2 μm.

Supplementary Fig. 3c) did not lead to an enhanced defect in parasite fitness (Supplementary Fig. 4d).

The perfect co-localization of TgCEN2 and TgMyoJ at the basal cup suggested that these two proteins might act together in basal constriction. Tet-repressive parasite line of TgCEN2 was generated using the same strategy as for MycUNC-iKD except

that no N-terminal tag was introduced (Supplementary Fig. 3d). CEN2-iKD was subsequently C-terminally tagged with YFP to follow its downregulation on addition of ATc. After 48 h of treatment, CEN2-YFP-iKD was not detectable by WB (Fig. 4d and Supplementary Fig. 11) and only slightly detectable as weak dots at the centrosome (Fig. 4e). Depletion of TgCEN2 caused an

enlarged posterior ring containing TgMyoJ without apparent defect in cytokinesis (Fig. 4f). As for MyoJ-KO, the same reduced constriction was measured using the endogenously tagged TgMyoC (Supplementary Fig. 2f). To assess the contribution of actin in this process, intracellular parasites expressing MyoJ-3Ty and CEN2-YFP were treated for 4 h with 1 μM of CytD. The diameter of the ring staining of TgMyoJ increased by twofold in treated parasites compared to the untreated ones (Fig. 4g). Moreover, the posterior staining of TgCEN2 was no longer detectable, either lost or diluted along the enlarged basal cup.

These findings establish that TgMyoJ and TgCEN2 act in concert to constrict the basal complex at the end of division in an actin-dependent manner.

**Impact of TgMyoI and TgMyoJ on residual body formation.** To assess the formation of RB in absence of a proper marker of this structure, we deleted the $Ca^{2+}$-dependent protein kinase TgCDPK2, known to lead to an imbalance between starch synthesis and degradation and a dramatic accumulation of amylopectin granules in RB that can be visualized by periodic acid–Schiff (PAS) staining[37]. *TgCDPK*2 was disrupted in MyoI-KO, MyoJ-KO and in ΔKU80 control strain without additional defect on the fitness of the parasites (Fig. 5a). In ΔKU80, amylopectin accumulated mainly in the RB at the centre of the rosettes as detected by PAS staining by differential interference contrast (DIC) (Fig. 5b), whereas it accumulated at the posterior but mainly inside the parasites in MyoI-KO/CDPK2-KO and MyoJ-KO/CDPK2-KO (Fig. 5b,c). TEM sections taken on intravacuolar parasites revealed that MyoI-KO/CDPK2-KO and MyoJ-KO/CDPK2-KO do actually form RB in which amylopectin accumulates but the parasites appear not to be connected to the RB in contrast to CDPK2-KO parasites (Fig. 5c).

**Cell-cell communication ensures synchronized division.** The loss of intravacuolar synchronized division observed in TgUNC-depleted parasites (Fig. 2l) was recapitulated in MyoI-KO and MyoJ-KO mutants based on detection of TgIMC1 and TgISP1 by IFA (Fig. 6a). The phenomenon becomes more pronounced as the number of division increases (Fig. 6b). In wild-type parasites, the loss of synchronicity was observed in large vacuoles (32 parasites/vacuole) when the rosette organization is compromised due to space constraint (Supplementary Fig. 5a). We reasoned that a connection must exist between tachyzoites within a vacuole, allowing soluble molecules such as cyclins or other cell cycle regulators to diffuse between parasites to tightly control the checkpoints of the cell cycle[17]. Consistently, a cell cycle regulated nuclear H4K20 methylase, TgSET8, which remains tightly regulated even when expressed ectopically[38] was examined in the absence of TgMyoI or TgMyoJ. In contrast to ΔKU80, where the signal of HA-SET8, when detected, was expressed with the same intensity in all parasites of a vacuole, in MyoI-KO and MyoJ-KO, HA-SET8 signal was only detectable in few parasites per vacuole confirming the intravacuolar desynchronization (Fig. 6c).

To assess the existence of cell–cell communication, fluorescence recovery after photobleaching (FRAP) experiments were performed on ΔKU80, MyoI-KO and MyoJ-KO parasites transiently transfected with soluble GFP. One or more parasites were selectively bleached and the intensity of fluorescence was recorded in bleached parasites as well as in the parasites residing within the same vacuole. In ΔKU80 parasites, the recovery of fluorescence was fast (about 1 min) and concomitant with a decrease of fluorescence in the neighbouring parasites indicating a diffusion of the GFP into the bleached parasite (Fig. 6d and Supplementary Movie 1). In contrast, the fluorescence in the

neighbouring vacuoles remained constant over the time of the experiment (Supplementary Fig. 5b and Supplementary Movie 1). The speed of fluorescence recovery is proportional to the number of fluorescent parasites connected (Supplementary Fig. 5b and Supplementary Movie 1). In large vacuoles (32 parasites/vacuole), the speed of fluorescence recovery is slower (Supplementary Fig. 5b and Supplementary Movie 2) and the connection is partially lost where the rosette organization is compromised (Supplementary Fig. 5b and Supplementary Movie 2). Importantly no fluorescence was recovered within 3 min when all the parasites were bleached within a vacuole, indicating that neo-synthesis of the GFP is not responsible for fluorescence recovery (Fig. 6e and Supplementary Movie 1). In sharp contrast, individually bleached MyoI-KO and MyoJ-KO parasites failed to recover fluorescence indicating that the parasites are no longer connected (Fig. 6f,g and Supplementary Movie 3). To confirm that synchronicity of division results from diffusion of soluble regulators that tightly controls the cell cycle progression, FRAP experiments have been performed on ΔKU80 parasites transiently transfected with GFP-SET8. When the signal of one nucleus of the vacuole was bleached, the recovery of fluorescence was observed within the following 5–20 min in contrast to the control vacuole confirming that the fluorescence recovery is due to diffusion and not to neo-synthesis of GFP-SET8 (Fig. 6h and Supplementary Movie 4). Of note, the cell–cell communication observed by FRAP was only intravacuolar since no recovery was observed between two independent vacuoles within the same host cell.

To exclude that the connectivity between intravacuolar parasites was an artifact of the *in vitro* culture, mice were infected with GFP-expressing parasites and peritoneal cells were collected 3–4 days later for FRAP analysis. Most of the vacuoles showed a recovery of fluorescence after bleaching proving the existence of intravacuolar connection between tachyzoites *in vivo* (Fig. 6i and Supplementary Movie 5).

Since the structure implicated in the connection is not comprehensively visible by IFA, TEM was used to image serial sections of wild-type parasites-containing vacuoles (Fig. 7a). As previously shown[13,39], intravacuolar tachyzoites appear to be connected at their basal pole via the residual body located at the centre of the rosette and share the same cytoplasm (Fig. 7b). TEM serial sections were used to perform 3D reconstruction of the vacuole (Fig. 7c,d). Strikingly, the 3D reconstruction revealed the presence of the tubular mitochondrion within the connection passing through the basal complex whose diameter measures around 322 nm (Fig. 7e,f and Supplementary Movies 6 and 7). This raised the possibility that diffusion or exchanges could occur between parasites through the mitochondrion. To test it, the matrix superoxide dismutase 2 fused to GFP (Nt-SOD2-GFP)[40] was used in the FRAP experiments; however, no recovery of fluorescence was recorded in the >10 vacuoles tested (Supplementary Movie 8).

Given the key participation of TgMyoI and TgMyoJ in the formation and/or the maintenance of the connection between parasites, the importance of actin dynamics was assessed. Parasites treated for 2 h with 1 μM of CytD were still connected and partial loss of connection was observed when parasites were treated for 8–10 h with a fluorescence recovery indicating that only a subset of parasites remained connected within a given vacuole (Supplementary Fig. 5c and Supplementary Movie 9).

Collectively, parasite connectivity allows diffusion of soluble molecules between intravacuolar parasites and hence ensures synchronized division. This cell–cell communication is dependent on TgMyoI and TgMyoJ. Only a long-term treatment with CytD that encompasses a complete division cycle impacts on connectivity suggesting the involvement of stable F-actin.

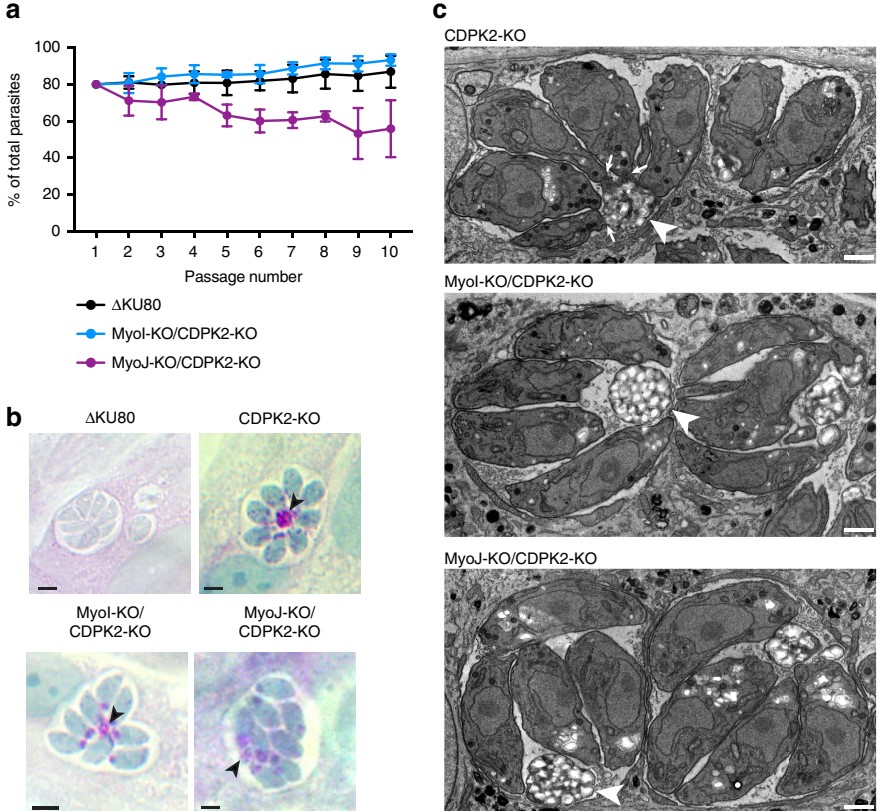

**Figure 5 | Deletion of TgMyoI or TgMyoJ does not impact on the residual body formation.** (**a**) The accumulation of amylopectin within the parasites in MyoI-KO/CDPK2-KO and MyoJ-KO/CDPK2-KO had no additional impact on their fitness as shown by competition assay using GFP-expressing parasites as internal control. Results are expressed as mean ± s.d. ($n = 3$). (**b**) PAS staining of ΔKU80, CDPK2-KO, MyoI-KO/CDPK2-KO and MyoJ-KO/CDPK2-KO parasites. In CDPK2-KO, amylopectin was mostly accumulated in the RB (arrowhead), whereas in MyoI-KO/CDPK2-KO and MyoJ-KO/CDPK2-KO, amylopectin was found in the RB (arrowheads) but also accumulating at the basal end of the parasites. Scale bar, 2 μm. (**c**) TEM sections of intracellular CDPK2-KO, MyoI-KO/CDPK2-KO and MyoJ-KO/CDPK2-KO parasites showing that one RB (arrowheads) is visible in each vacuole but not connected (arrows) to the parasites in MyoI-KO/CDPK2-KO and MyoJ-KO/CDPK2-KO.

**Cell–cell communication and delayed death phenotype.** The continuity between parasites allows the diffusion of soluble proteins and plausibly a large range of metabolites and even small vesicles. In this context, we examined the phenomenon of DDP, originally described in *T. gondii* tachyzoites, treated with clindamycin, ciprofloxacin or chloramphenicol[15,16,41]. DDP is associated with the loss of apicoplast inheritance, which can be triggered by expression of dominant-negative mutants such as ACP-GFP-mROP1 (ref. 16) or DD-MyoF-tail[10]. We reasoned that these mutants survived during the first lytic cycle due to the connection between parasites enabling the diffusion of one or more apicoplast-derived metabolites that can fuel all the parasites of the vacuole even if only few of them retained a functional apicoplast. To test this hypothesis, *TgMyoI* and *TgMyoJ* were individually deleted in parasites expressing stably DD-MyoF-tail on addition of shield (Shld-1)[10] (Supplementary Fig. 6). The impact of DD-MyoF-tail expression on the various strains was assessed over 72 h and GFP-expressing parasites were mixed as an internal control (Fig. 8a). All the parasites grew like the GFP parasites in the absence of Shld-1. In contrast, the deleterious effect of DD-MyoF-tail expression was observed in the second lytic cycle in wild-type parasites but was already prominent in the first lytic cycle in DD-MyoF-tail/MyoI-KO and DD-MyoF-tail/MyoJ-KO parasites. Collectively, cell–cell communication confers resistance to the parasites lacking the apicoplast during the first lytic cycle and hence participates in the DDP (Fig. 8b).

**Cell–cell communication is absent in mature tissue cysts.** The existence of an intravacuolar connection between parasites is not only a hallmark of the type I virulent strain RH since the observations were reproduced in the cyst-forming type II strain ME49 where both *TgMyoI* and *TgMyoJ* genes were disrupted (Supplementary Fig. 7a,b). Like in type I, type II ME49 MyoJ-KO tachyzoites formed slightly smaller lysis plaques after 10 days (Supplementary Fig. 7c) and exhibited a loss of fitness compared to wild-type and ME49 MyoI-KO parasites (Fig. 9a). While wild-type ME49 parasites exhibit a basal level of asynchronicity of division in about 15% of the vacuoles, this raised to more than 50% when TgMyoI or TgMyoJ were deleted and this asynchronicity increased with the number of divisions (Fig. 9b). Concordantly, FRAP experiments performed on GFP-expressing parasites confirmed that ME49 tachyzoites were connected in a TgMyoI- and TgMyoJ-dependent manner (Supplementary Fig. 7d–f and Supplementary Movie 10).

During natural infection, tachyzoites rapidly encounter resident macrophages, dendritic cells and intraepithelial lymphocytes. Accordingly, we analysed cell–cell communication of both RH and ME49 tachyzoites in activated and non-activated bone marrow-derived macrophages (BMDMs). While RH parasites were largely connected (Fig. 9c, top panel and Supplementary Movie 11), ME49 parasites were only connected in non-activated BMDMs while communication was lost in activated BMDMs (Fig. 9c, bottom panel, Fig. 9d and

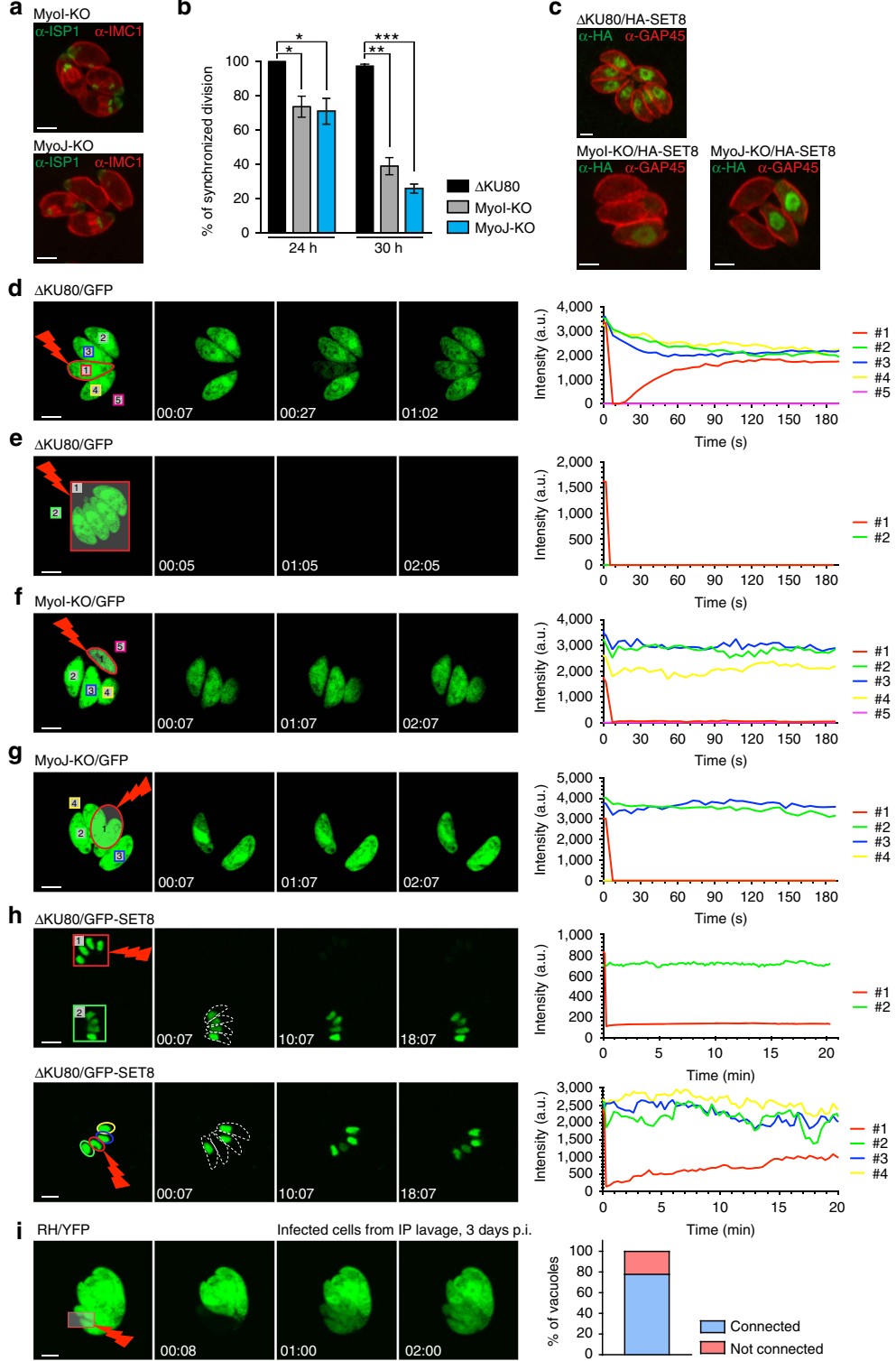

**Figure 6 | Intravacuolar TgMyoI-KO and TgMyoJ-KO parasites divide asynchronously and are not connected.** (**a**) IFA showing the intravacuolar asynchronicity of division in MyoI- and MyoJ-KO stained with α-ISP1 and α-IMC1 24 h post infection. Scale bar, 2 μm. (**b**) Quantification of the synchronicity of intravacuolar division after 24 and 30 h of growth. Results are represented as mean ± s.d. from three independent experiments and the significance of the results was assessed using a parametric paired *t*-test. The two-tailed *P*-values are: 0.0219 (*) and 0.0175 (*) for MyoI-KO and MyoJ-KO, respectively, compared to ΔKU80 at 24 h and 0.0016 (**) and 0.0007 (***) at 30 h. (**c**) Transient transfection of SET8-HA shows that intravacuolar parasites are all in the same phase of their cell cycle in ΔKU80 in contrast to intravacuolar MyoI- and MyoJ-KO parasites. Scale bar, 2 μm. (**d–h**) Left panels: time-lapse imaging of FRAP experiments performed on type I tachyzoites ΔKU80 (**d,e**), MyoI-KO (**f**) and MyoJ-KO (**g**) transiently transfected with GFP or on type I tachyzoites ΔKU80 transiently transfected with SET8-GFP (**h**). The bleached areas are delineated in red. Right panels: quantification of the intensity of GFP fluorescence recorded in the areas numbered on the left panels. Scale bars, 2 μm (**d–g**) and 5 μm (**h**). (**i**) Time-lapse imaging of a representative FRAP experiment performed on intracellular type I tachyzoites (RH-YFP) collected by intraperitoneal (IP) lavage 3 days p.i. (scale bar, 2 μm) and quantification of parasites connection in 10 vacuoles tested.

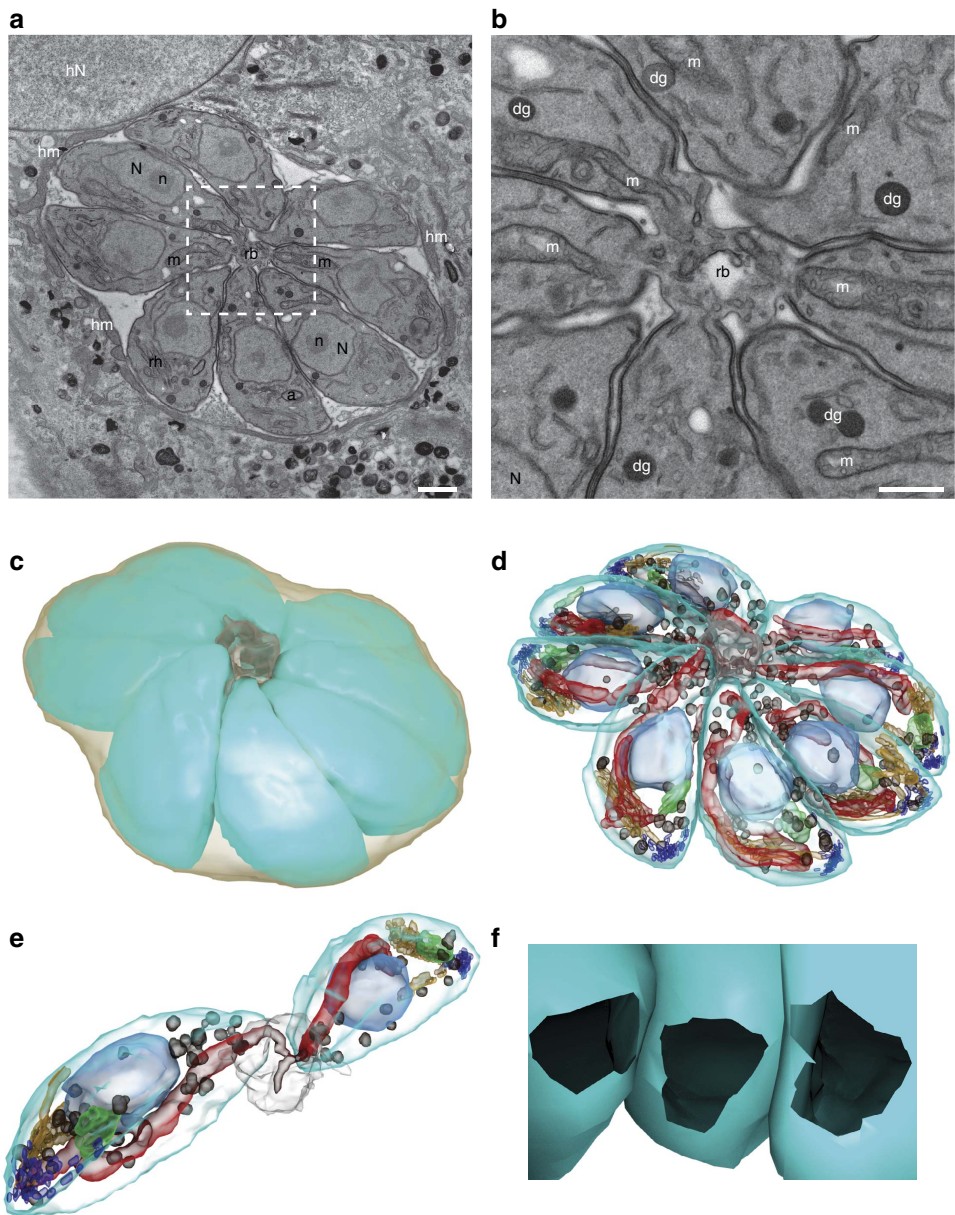

**Figure 7 | ΔKU80 parasite intravacuolar connection revealed by serial section TEM and 3D reconstruction.** (**a**) Section from the imaged volume through the PV with eight tachyzoites connected to the residual body (rb) outlined by white dashed square. The PV inside the host cell cytoplasm is surrounded by host mitochondria (hm) and located close to the host cell nucleus (hN). Scale bar, 1 μm. (**b**) High magnification view of the centre of the rosette. The tachyzoite organelles nucleus (N) with nucleolus (n), mitochondrion (m), rhoptry (rh), apicoplast (a) and dense granules (dg) are present. Scale bar, 0.5 μm. (**c,d**) 3D model of the reconstructed rosette from 33 serial sections of total imaged volume of 18.14 μm × 18.14 μm × 2.31 μm. (**c**) The individual tachyzoites (cyan) forming the rosette are connected to the residual body (grey) inside the PV (light brown). (**d**) Same model showing the tachyzoite organelles: nucleus (light blue), mitochondrion (red), rhoptry (orange), apicoplast (green), micronemes (dark blue) and dense granules (dark brown). (**e**) View of two isolated tachyzoites connected to the residual body with closely apposed mitochondria (red). (**f**) View from the centre of the residual body showing the connection opening at basal pole of three tachyzoites. The diameter of this opening has been measured (n = 111) on several sections of three vacuoles and is estimated at 322 ± 53 nm (mean ± s.d.).

Supplementary Movie 11). The presence of a cyst wall marker, the *Dolichos biflorus* agglutinin (DBA) lectin was assessed under this experimental condition. While vacuoles formed by RH parasites are known not to be able to undergo stage conversion, ME49 vacuoles presented a basal level of DBA lectin staining both in activated and non-activated infected BMDMs (Fig. 9e). These observations suggest that the connection between parasites is dynamic and rapidly lost in parasite exposed to stress.

*In vitro* differentiated bradyzoites were reported to be asynchronous[42] suggesting that bradyzoites are not connected. To investigate

this further, ME49 tachyzoites were cultivated *in vitro* in conditions triggering differentiation into bradyzoites and maintained for several days to allow cyst formation as previously described[43]. The efficiency of stage conversion was assessed by IFA using α-P21 antibodies[44] and the DBA lectin (Supplementary Fig. 8), and the proper formation of the cyst wall was verified by TEM (Fig. 10a). In bradyzoites differentiated *in vitro* during 14 days, we always observed asynchronicity of division with single parasite dividing within the cyst (Fig. 10b). FRAP experiments showed that in early cysts (from day 1 to day 6), parasites are connected two by two,

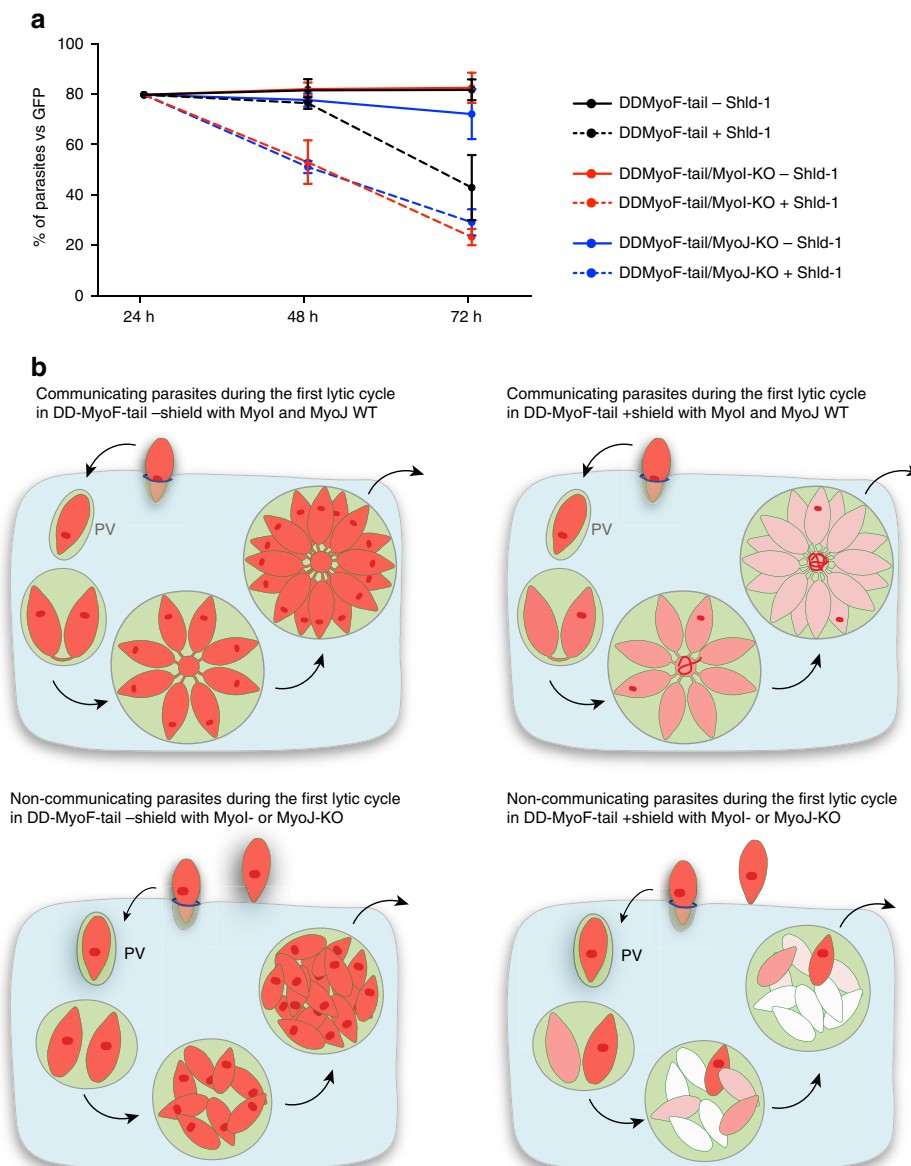

**Figure 8 | The intravacuolar connection contributes to the delayed death phenotype.** (**a**) The survival of the DD-MyoF-tail, DD-MyoF-tail/MyoI-KO and DD-MyoF-tail/MyoJ-KO parasites was assessed at 24, 48 and 72 h p.i. in absence or presence of Shield-1 (Shld-1) using GFP-expressing parasites as an internal control. For comparison, the ratios of GFP control parasites at 24 h have been normalized to 20%. The results are expressed as mean ± s.d. ($n=3$). (**b**) Model of parasites' growth during the first lytic cycle depending on the presence or absence of TgMyoI and TgMyoJ and the defect ( + Shld-1) or not ( − Shld-1) in apicoplast segregation. The diffusion of metabolites from the apicoplast (red dot) is represented in red. The different shades of red reflect the dilution of metabolites occurring on division when the apicoplast is not properly segregated.

whereas after day 6, bradyzoites became progressively disconnected (Fig. 10c, Supplementary Fig. 9 and Supplementary Movies 12 and 13). In contrast, ME49 MyoI-KO and ME49 MyoJ-KO bradyzoites switched *in vitro* never appeared connected. These *in vitro* observations were confirmed on mature cysts collected from the brain of mice chronically infected with ME49-expressing GFP. Relevantly, the cyst-forming bradyzoite stage of *T. gondii* was reported to be less quiescent than previously assumed with isolated or patches of parasites often seen to divide asynchronously within cysts *in vivo*[45]. The FRAP experiments performed on purified cysts showed no connection between bradyzoites even two by two (Fig. 10d and Supplementary Movie 14).

The importance of TgMyoI and TgMyoJ as virulence factors *in vivo* was assessed by intraperitoneal infection of mice. Infection with either 500 or 1,000 tachyzoites of the type II ME49 wild-type

or ME49 MyoI-KO strains led to the death of the animals with the same time frame attesting that MyoI-KO parasites are not impaired in virulence (Fig. 10e). In contrast, all the mice infected with type II ME49 MyoJ-KO survived infection without any apparent symptoms and fewer tissue cysts were detected from the brain of these animals compared to the control group and ME49 MyoI-KO parasites (Fig. 10f). The loss of virulence might result from a decreased mechanical resistance of MyoJ-KO compared to wild type or MyoI-KO as observed when these parasites were subjected to a hypo-osmotic shock (Supplementary Fig. 10). Yet, the survival is apparently not dose-dependent since mice also survived infection with 150,000 parasites of the ME49 MyoJ-KO strain. Taken together, cell–cell communication is maintained in fast-replicating tachyzoites of both the type I and type II strains but is rapidly lost in the slow-growing bradyzoites.

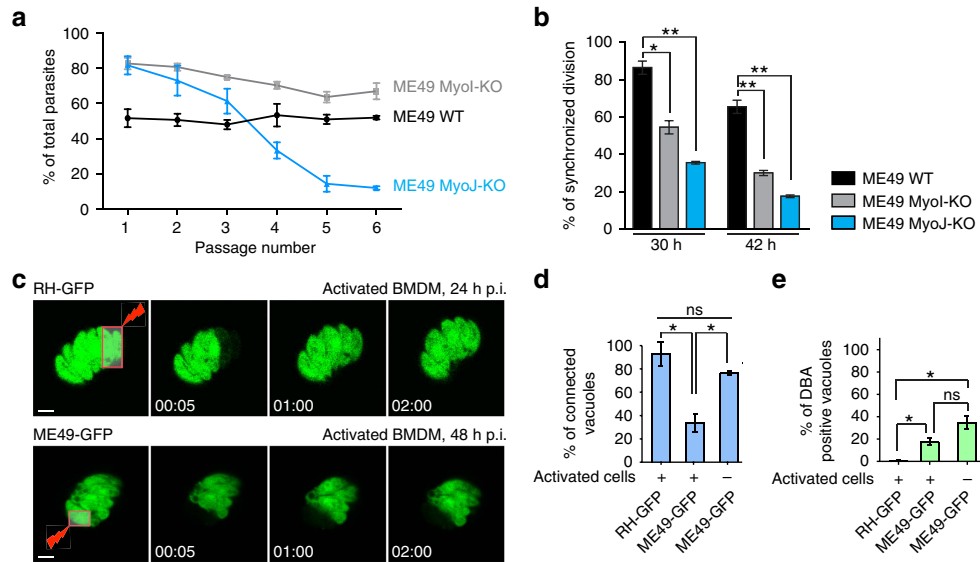

**Figure 9 | The connectivity between type II parasites shows higher susceptibility to stress conditions than type I.** (**a**) Competition assay performed over six passages with type II ME49 wild type (WT), MyoI-KO and MyoJ-KO using GFP-expressing parasites as an internal control. As in type I, type II MyoJ-KO parasites exhibit a fitness defect. The results are expressed as mean ± s.d. ($n = 3$). (**b**) Quantification of the synchronicity of intravacuolar division 30 or 42 h p.i. Results are represented as mean ± s.d. ($n = 3$) and their significance was assessed using a parametric paired *t*-test. The two-tailed *P*-values are: 0.0120 (\*) and 0.0025 (\*\*) for MyoI-KO and MyoJ-KO, respectively, compared to WT at 30 h and 0.0057 (\*\*) and 0.0028 (\*\*) at 42 h. (**c**) Time-lapse imaging of representative FRAP experiment performed activated BMDMs, 24 h after infection by type I (RH) tachyzoites (top panel) or 48 h after infection by type II (ME49) tachyzoites (bottom panel), all stably expressing GFP. Scale bar, 2 μm. (**d**) Quantification of parasites connection in *n* different vacuoles recorded from two biological replicates: $n = 15$ (RH, first column), $n = 36$ (ME49, second column), $n = 26$ (ME49, third column). Results are presented as mean ± s.d. and their significance was assessed using a parametric paired *t*-test. The two-tailed *P*-values are: 0.0224 (left \*) and 0.0172 (right \*). (**e**) Quantification of tachyzoite to bradyzoite conversion assessed by DBA lectin staining of RH-GFP (24 h p.i.) or ME49 (48 h p.i.) infected BMDMs, in 100 vacuoles from two biological replicates. Results are presented as mean ± s.d. and their significance was assessed using a parametric paired *t*-test. The two-tailed *P*-values are: 0.0168 (left \*) and 0.0150 (right \*).

## Discussion

*T. gondii* possesses the largest repertoire of myosin motors among the members of Apicomplexa[2,46]. TgUNC belongs to the UCS family of myosin chaperones and is critical for the assembly of all members of the five classes of *Toxoplasma* myosin heavy chains. TgUNC depletion recapitulates the phenome of myosins implicated in motility, invasion and egress[7,9]. These crucial steps of the lytic cycle rely on the concerted action of TgMyoH, TgMyoA and TgMyoC and no additional motor participating in motility was uncovered here, although functional redundancy or compensatory mechanisms cannot be excluded. The defect in apicoplast inheritance previously assigned to TgMyoF was only modestly reproduced in the absence of TgUNC, probably due to the incomplete disappearance of TgMyoF. Importantly, no myosin appears to play a critical role in parasite division as was previously but only indirectly deduced from CytD treatment[47]. Moreover, myosins do not seem to be implicated in conoid protrusion despite a previous report describing it as an actomyosin-dependent process[30]. However, the roles of TgMyoE and TgMyoL at the conoid remain unknown.

Importantly, two novel phenotypes have been uncovered on depletion of TgUNC. The process of basal complex constriction was assigned to TgMyoJ, a class XXIII (VI-like) myosin present in all Apicomplexans with the exception of *Theileria* and *Babesia* spp.[2,46]. In the absence of TgMyoJ, the parasites fail to constrict their basal complex although cytokinesis occurs normally. Strikingly, an enlargement of the basal complex was measured on treatment with CytD although previous studies did not report this defect[33,47] and hence was inappropriately assumed to be an actomyosin-independent process[34]. TgCEN2 co-localizes with TgMyoJ at the basal cup and is absent in MyoJ-KO. Depletion of TgCEN2 prevents the constriction of the basal pole without

affecting the localization of TgMyoJ. Since TgCEN2 is a small EF-hand-containing protein, it could conceivably act as myosin light chain for TgMyoJ and modulate its function to generate the contractile force or alternatively act as cargo and be brought to the site of action by TgMyoJ. Despite this spectacular morphological phenotype, the apparently 'truncated' parasites survived albeit with a loss of fitness.

Following invasion and sealing of the PVM, *T. gondii* tachyzoites divide asexually by a process called endodyogeny and implying that the two daughter cells are built inside the mother parasite[17]. At the end of the division, the daughter cells separate at their posterior pole but appear to remain connected via the RB that constitutes a vacuolar network which maintains the parasites spatially organized in rosettes within the PV during the subsequent rounds of division[13]. The RB varies in size depending on parasite fitness and especially on its ability to undergo proper division with accurate positioning of the organelles. Typically, enlarged RBs were previously observed in parasite treated with drugs disrupting actin polymerization[47], on perturbation of TgMyoF function[10] or when illegitimate amylopectin granules are formed on disruption of *TgCDPK2* (ref. 37). In the absence of TgMyoI or TgMyoJ, parasites do not form a RB except when forced to do so by deletion of *TgCDPK2*. This result is in concordance with the fact that the dramatic accumulation of mislocalized organelles in the RB monitored in absence of TgMyoF[10] was not recapitulated in parasites depleted in TgUNC. TgMyoI, a class XXIV myosin[2], is the first reported marker of this vacuolar compartment and is mislocalized in MyoJ-KO parasites. This plausibly explains the dual phenotype observed in absence of TgMyoJ and points towards TgMyoI being the motor responsible for the formation/maintenance of the connection. The RB contributes along with the orientation of the

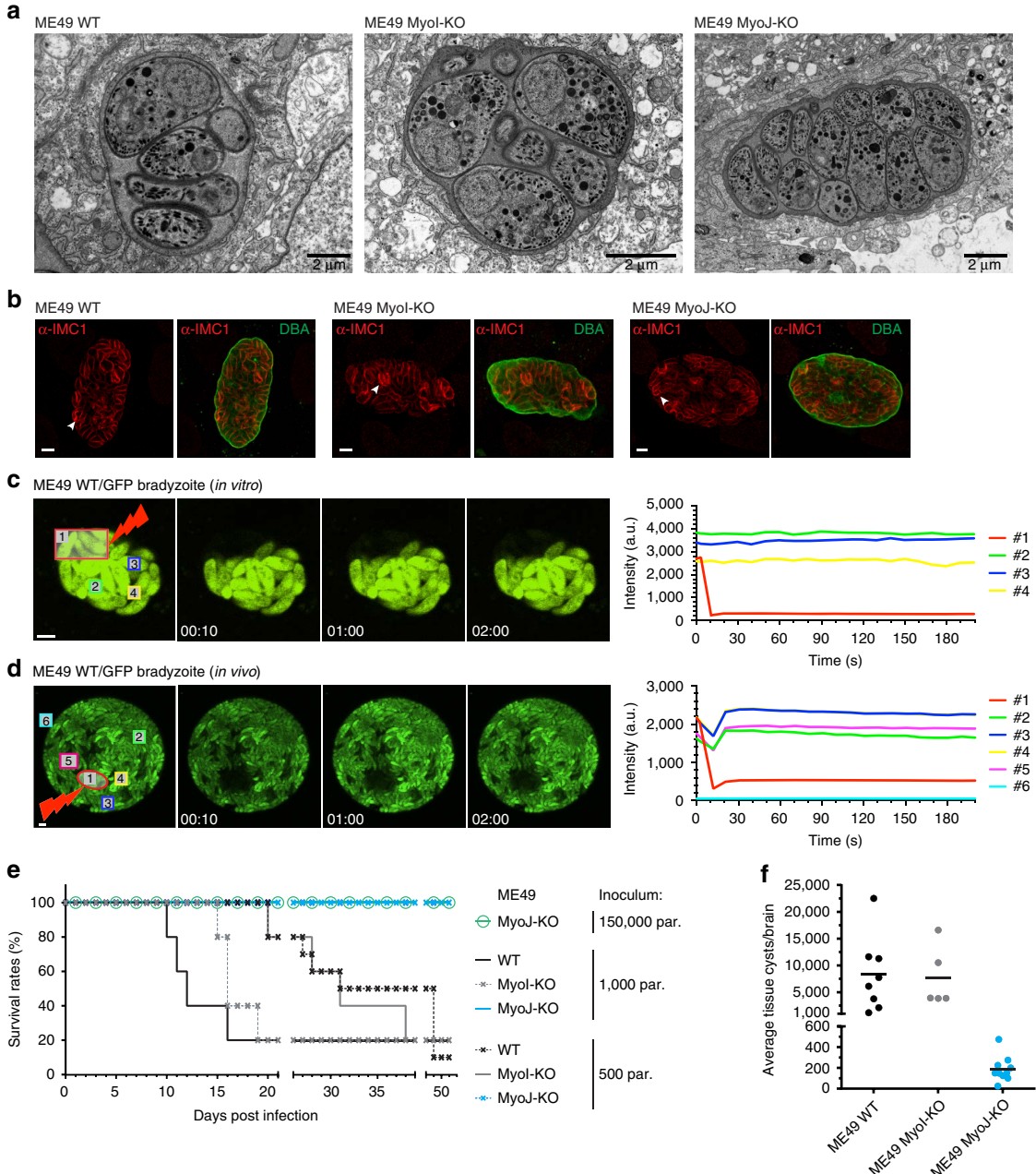

**Figure 10 | The connectivity between bradyzoites is lost both *in vitro* and *in vivo*.** (**a**) Electron micrographs of cyst-forming bradyzoites 14 days post-differentiation *in vitro*. No visible defect was detected in MyoI-KO or MyoJ-KO compared to WT. Scale bar, 2 μm. (**b**) The asynchronicity of division within cysts 14 days post-differentiation *in vitro* is visible in the MyoI-KO and MyoJ-KO but also in the WT strain using α-IMC1 (arrowheads). Scale bar, 5 μm. (**c,d**) Left panels: time-lapse imaging of FRAP experiments performed on WT bradyzoites 14 days after differentiation *in vitro* (**c**) or collected from mouse brain 4 weeks p.i (**d**). The bleached areas are delineated in red. Scale bar, 5 μm. Right panel: quantification of the intensity of GFP fluorescence recorded in the areas numbered on the left panel. (**e**) 500 or 1,000 ME49 tachyzoites of the parental (WT) or knockouts were injected intraperitoneally into CBA/J mice. In addition, 150,000 parasites of the ME49 MyoJ-KO strain have also been injected. Five animals were infected for each strain and their survival was monitored over time. (**f**) Average number of cysts per brain assessed after tissue cyst extraction.

daughter cells to the arrangement in rosette of the tachyzoites within the PV. The TEM-3D reconstruction of a vacuole allowed the visualization of the structure responsible for the connection and confirmed that intravacuolar parasites indeed share the same cytosol. This connection and concomitantly the synchronized division are lost in absence of TgMyoI or TgMyoJ. The FRAP experiments clearly demonstrate the existence of a cell–cell communication between intravacuolar tachyzoites that allows the diffusion of soluble cytosolic GFP as well as the nuclear cell cycle regulated H4K20 methylase TgSET8. The diffusion of GFP and

TgSET8 strongly suggests that all sorts of soluble metabolites can diffuse between parasites, including metabolites produced by the apicoplast and postulated to participate in the DDP phenomenon. Concordantly, parasites lacking the apicoplast as consequence of impairment of TgMyoF function are more severely affected during the first lytic cycle if they lack either TgMyoI or TgMyoJ. Similarly, in *Plasmodium*, the DDP could simply be explained by the shared cytoplasm allowing the diffusion of metabolites until the parasites segmented at the very last step of schizogony. The presence at high frequency of the tubular mitochondrion in the

connection as observed by TEM serial sections suggests possible exchanges between parasites through this organelle. However, this does not occur at the level of matrix protein as shown with SOD2-GFP. Interestingly, in mammalian cells, mitochondria have been observed within nanotubular cell-to-cell connections termed membrane nanotubes enabling the exchange of material, energy, but also signals between cells[48] and in analogy, most of the membrane nanotubes contain actin filaments.

Importantly, the connectivity in *T. gondii* is observed between intravacuolar tachyzoites of both type I (RH) and type II (ME49) strains and is not maintained during development of the cyst-forming bradyzoites. Specifically, type II strain is sensitive to stress stimuli and prone to lose connection when exposed to activated macrophages. In this context, disconnected parasites might be preparing for bradyzoite differentiation. The FRAP experiments performed on advanced *in vitro* differentiated parasites as well as on *in vivo* tissue cyst show that bradyzoites are clearly not connected. This finding correlates with the recent observation of low level of synchronized division of bradyzoites within cysts purified from the central nervous system of mice[45]. Interestingly, during the initial cycles of bradyzoite division, the parasites remained connected two by two, which explains why TgMyoI is expressed in bradyzoite stage. It is plausible that the link established by TgMyoI at the end of the endodyogeny process is actively broken to slow down the growth of the cyst and participate in the process of latency during chronic infection.

The lack of connection observed in type II MyoI-KO tachyzoites had no impact on virulence compared to wild type in the mice model. In contrast, type II ME49 MyoJ-KO are unable to kill the mice even at relatively high doses. Concordantly very few cysts were detectable in the brain of mice infected with ME49 MyoJ-KO parasites although they appeared morphologically normal. The reduced fitness possibly attributed to loss of mechanical resistance observed in the competition assays and also reported in the recent CRISPR/Cas9 screen of the *T. gondii* genome[49] might explain the phenomenon although further investigation is needed to unravel the nature of this lack of virulence.

The implication of a myosin in the connectivity between two cells raises a number of challenging questions about the nature of the exchange. The enlarged basal pole and asynchronous division can be also observed in TgACT1-iKO parasites[50]. In this study, chromobodies that specifically bind F-actin, detected filaments in the RB and emanating from the basal pole of parasites confirming the involvement of the actomyosin system in parasite connectivity and communication. Finally, a key step of the lytic cycle is the egress from infected cells with the concomitant switch of parasites to the motility mode. Individualization of the parasites appears to be a prerequisite for survival and spreading and thus the RB must collapse with proper sealing of the parasites at the time of egress. In consequence, signalling cascade leading to microneme secretion, motility and parasites sealing must operate in a tight, timely concerted fashion, for example by responding simultaneously to signalling molecules such as $Ca^{2+}$ and/or cGMP.

## Methods

**Preparation of *T. gondii* genomic DNA and RNA.** Genomic DNA (gDNA) has been prepared from tachyzoites (RH and ME49 strains obtained from aidsreagent.org and ATCC, numbers are 50174 and 50611) using the Wizard SV genomic DNA purification system (Promega).

**Cloning of DNA constructs.** All amplifications for cloning were performed with the LA Taq (TaKaRa) or Q5 (New England Biolabs) polymerases and the primers used are listed in Supplementary Table 1. All amplifications for screening were performed with the GoTaq DNA polymerase (Promega) and the primers are listed in Supplementary Table 2.

To introduce epitope tag to myosins, their endogenous loci were modified with 'knock-in' vectors. The gDNA fragment of the C-terminal part of *TgMyoE* (TGME49_239560), *TgMyoG* (TGME49_314780), *TgMyoI* (TGME49_230980), *TgMyoJ* (TGME49_257470) and *TgMyoK* (TGME49_206415) was amplified by PCR using the primers listed in Supplementary Table 1, digested with the restriction enzymes listed in Supplementary Table 1 and cloned into *Kpn*I or *Mfe*I and *Nsi*I sites of the pTUB8MIC13-3Ty-HX (ref. 51). Before transfection, the plasmids have been linearized in the middle of the cloned gDNA fragment. For knock-in insertion of these vectors into the UNC-iKD strain, the HXGPRT cassette has been exchanged with a DHFR-TS cassette subcloned into the two *Sac*II sites. For *TgMyoL* (TGME49_291020), a gDNA fragment has been amplified using the primers TgMyoL-5549/TgMyoL-5548, digested with *Apa*I and *Sbf*I and cloned into the *Apa*I and *Nsi*I sites of the Ct-ASP5-3Ty-DHFR[52]. All vectors were transfected in RHΔKU80 (ref. 53).

The myosins of interest have been knocked out in the RHΔKU80 strain using a knock-in strategy recombining into the head domain thus creating a truncated and non-functional protein. A gDNA fragment of the head domain of TgMyoE, TgMyoG, TgMyoI, TgMyoJ, TgMyoK and TgMyoL was amplified by PCR using the primers listed in Supplementary Table 1 and inserted into the *Kpn*I and *Nsi*I restriction sites of the pTUB8MIC13-3Ty-HX (ref. 51).

To generate TgMyoI-KO and TgMyoJ-KO in the type II ME49ΔHX strain, the CRISPR-Cas9 system was used to disrupt the genes by insertion of a DHFR-TS selection cassette into the coding sequence of the head domain. To do so, a PCR product was generated using the KOD DNA polymerase (Novagen, Merk) with the vector p2854-DHRF (ref. 54) as template and the primers TgMyoI-6022/TgMyoI-6023 or TgMyoJ-6025/TgMyoJ-6026 that also carry 30 bp homology with the respective myosin locus. To direct the insertion of this PCR product, a specific gRNA vector has been generated for each myosin using the Q5 site-directed mutagenesis kit (New England Biolabs) and the vector pSAG1::CAS9-GFP-U6::sgUPRT as template[55]. The UPRT-targeting gRNA was replaced by a TgMyoI- or TgMyoJ-specific gRNA using the primer pair TgMyoI-6021/gRNA-4883 and TgMyoJ-6024/gRNA-4883, respectively. To generate TgMyoI-KO and TgMyoJ-KO in the DD-MyoF-tail strain, the same specific sgRNA vectors were used with the CRISPR-Cas9 system to disrupt the genes. Parasites expressing the Cas9-YFP were sorted by flow cytometry (FACS) and cloned into 96-well plates using a MoFlo Astrios (Beckman Coulter). The clones were then analysed by sequencing.

To generate the UNC-3Ty-LoxP-3′UTR-LoxP-U1 vector, a gDNA fragment corresponding to the C terminus of *TgUNC* (TGME49_249480) was amplified by PCR using the primers TgUNC-5076/TgUNC-5077 and cloned into the *Kpn*I and *Nsi*I sites of the pG152-3Ty-LoxP-3′UTRSag1-HXGPRT-LoxP-U1 plasmid[26] previously modified to introduce a unique *Kpn*I site and a sequence coding for a 3xTy-tag.

To generate tet-repressive knockdown of TgUNC, a PCR fragment encoding the TATi trans-activator, the HXGPRT cassette and the TetO7S1 promoter was generated using the KOD DNA polymerase (Novagen, Merk) with the vector 5′MyoF-TATi1-HX-tetO7S1MycNtMyoF[10] as template and the primers TgUNC-5364/TgUNC-5365 that also carry 30 bp homology with the 5′ end of *TgUNC*. To direct the insertion of the PCR product at the start of *TgUNC*, a specific sgRNA vector has been generated as described above using the primer pair TgUNC-5366/gRNA-4883.

MORN1 was tagged using the CRISPR-Cas9 strategy to introduce a MycGFP-tag at the N terminus of the endogenous gene. To do so, a PCR product was generated using the vector pTub8MycGFPPfMyoAtailTy-HX[56] as template and the primers TgMORN1-5609/TgMORN1-5610 that also carry 30 bp homology with the 5′ end of *TgMORN1*. To direct the insertion of this PCR product, a specific sgRNA vector has been generated as described above using the primer pair TgMORN1-5608/gRNA-4883.

To generate TgUNC-expressing constructs for complementation experiments, the coding sequences of *TgUNC* and *TgUNCΔTPR* were amplified from gDNA using the primer pairs TgUNC-5689/TgUNC-5077 and TgUNC-5690/TgUNC-5077, respectively, and then digested with *Mfe*I and *Nsi*I for cloning into the *Eco*RI and *Nsi*I sites of the pTUB8MycGFPPfMyoAtailTy-HX vector[56]. These plasmids were linearized with *Bsr*GI before transfection.

To generate TgCDPK2-KO in ΔKU80, TgMyoI-KO and TgMyoJ-KO backgrounds, the CRISPR-Cas9 system was used to disrupt the gene by insertion of a DHFR-TS selection cassette at the beginning of the gene. To do so, a PCR product was generated using the vector p2854-DHFR (ref. 54) as template and the primers TgCDPK2-6019/TgCDPK2-6020 that also carry 30 bp homology with the locus. To direct the insertion of this PCR product, a specific sgRNA vector has been generated as described above using the primer pair TgCDPK2-6016/gRNA-4883.

To generate tet-repressive knockdown of TgCEN2, a PCR fragment encoding the TATi trans-activator, the HXGPRT cassette and the TetO7S1 promoter was generated using the KOD DNA polymerase (Novagen, Merk) with the vector 5′MyoF-TATi1-HX-tetO7S1MycNtMyoF[10] as template and the primers TgCEN2-6314/TgCEN2-6315 that also carry 30 bp homology with the 5′ end of *TgCEN2*. To direct the insertion of the PCR product at the start of *TgCEN2* locus, a specific sgRNA vector has been generated as described above using the primer pair TgCEN2-6313/gRNA-4883. *TgCEN2-iKD* was then endogenously C terminally tagged with YFP using the pLIC-CEN2-YFP-CAT vector[57] linearized with *Eco*RV.

To generate GFP-SET8 expression plasmid, the GFP sequence from pT8-MycGFPPfMyoAtail-Ty[56] was amplified using primers SET8-6989 and SET8-6990, digested EcoRI/BamHI and cloned into the pGRA1-HATgSET8ΔN1 vector[38] digested with the same enzymes.

**Toxoplasma gondii tachyzoite and bradyzoite cultures.** *T. gondii* tachyzoites strains and their derivatives expressing the epitope-tagged proteins were grown in confluent human foreskin fibroblasts (HFF) maintained in Dulbecco's modified eagle's medium (Gibco DMEM) supplemented with 5% foetal calf serum, 2 mM glutamine and 25 μg ml$^{-1}$ gentamicin. Conditional expression of the Tet-inducible constructs was performed using 1 μg ml$^{-1}$ ATc[3].

The *in vitro* conversion from tachyzoite to bradyzoite was induced by replacing normal media with RPMI 1640 with 50 mM HEPES to pH 8.2 and supplemented with 3% foetal bovine serum. Parasites were allowed to grow at 37 °C in absence of $CO_2$ for multiple days and alkaline media was changed daily.

**Parasite transfection and selection of stable transformants.** Parasite transfections were performed by electroporation as previously described[58]. RHΔKU80 (ref. 53) and derivative strains have been transfected with 15–20 μg of the knock-in constructs. Either mycophenolic acid (MPA, 25 mg ml$^{-1}$) and xanthine (50 mg ml$^{-1}$) or pyrimethamine (1 μg ml$^{-1}$) was used to select the resistant parasites carrying the HXGPRT or the DHFR cassette, respectively. For the CRISPR-Cas9 strategy, 15 μg of the gRNA-specific CRISPR/CAS9 vector was transfected into RHΔKU80 together with the product of two PCR reactions and selected according to the selection cassette used. UNC-iKD strain was complemented with 60 μg of linearized pTub8-UNC-Ty-HX or pTub8-UNCΔTPR-Ty-HX under ATc selection.

The *hxgprt* locus has been disrupted into the type II ME49 strain using the CRISPR-Cas9 system and a specific sgRNA generated as previously described with the primer pair gRNA-HX-5950/gRNA-4883. Parasites expressing the Cas9-YFP were sorted by flow cytometry (FACS) and the clones were analysed by sequencing (Supplementary Fig. 6a). In this background, TgMyoI was endogenously tagged using the pKI-MyoI-3Ty-HXGPRT construct described above and in the same background as well as in the derived MyoI-KO and MyoJ-KO, the pTub-GFP-HXGPRT construct[56] was stably integrated after *Not*I linearization.

**Antibodies.** The antibodies used in this study were previously described as follow: polyclonal rabbit: α-MyoA and α-MLC1 (ref. 59), α-GAP40, α-IMC1 and α-GAP45 (ref. 9), α-MLC2 (ref. 28), α-Cpn60 (ref. 32), α-catalase[60], α-ARO[12], α-HSP70 (ref. 40); monoclonal mouse, α-ACT[59], α-ISP1 (ref. 61), α-Ty (BB2, ref. 62). α-SAG1, α-MIC2, α-GRA1, α-GRA3 and α-P21 are generous gifts from Dr J.-F. Dubremetz. For WB, secondary peroxidase-conjugated goat α-rabbit/mouse antibodies (Sigma) were used, as well as α-Myc (9E10, Santa Cruz Sc-40). For IFA, the secondary antibodies Alexa Fluor 488- and Alexa Fluor 594-conjugated goat α-mouse/rabbit antibodies (Life Technologies) as well as the DBA-fluorescein labelled (Reactolab FL-1031) were used. The dilutions of use are described in Supplementary Table 3.

**IFA.** Parasite-infected HFF cells seeded on coverslips were fixed with 4% paraformaldehyde (PFA) or 4% PFA/0.05% glutaraldehyde (PFA/GA) in PBS, depending of the antigen to be labelled. Fixed cells were then processed as previously described[56]. Confocal images were generated with a Zeiss LSM700 or LSM800 laser scanning confocal microscope using an apochromat × 63/1.4 oil objective at the Bioimaging core facility of the Faculty of Medicine, University of Geneva. Stacks of sections were processed with ImageJ and projected using the maximum projection tool.

**FRAP.** For FRAP experiments performed on tachyzoites, the pTub-GFP-HXGPRT[56], pTub-SOD2-Nterm-GFP-Myc (Nt-SOD2-GFP)[40] and GFP-HA-SET8 plasmids were transiently transfected into the parasites and the experiments were performed 24 h later for the RH strains or 48 h later for the ME49 strains. For the experiments performed on bradyzoites, ME49 strains stably expressing GFP were used and bradyzoite stage conversion was induced for 1–15 days. The experiments were performed on a Nikon A1r microscope (Ti Eclipse) controlled in temperature and $CO_2$. Acquisitions and processing were done with the softwares NIS-elements advanced research (Nikon) and ImageJ. The sequence used for the experiments was composed of an acquisition step of three images followed by two bleaches performed with 100% of laser (wavelength 488) and another acquisition step of 3–15 min with images collected every 5 or 10 s. All experiments were done at least in triplicates and 5–10 vacuoles were bleached each time.

All experiments were performed at the Bioimaging core facility of the Faculty of Medicine, University of Geneva.

**WB analysis.** Parasites were lysed in RIPA buffer (150 mM NaCl, 1% Triton X-100, 0.5% deoxycholate, 0.1% SDS, 50 mM Tris pH 7.5), incubated on ice for 15 min and then centrifuge for 30 min at 14,000 r.p.m. at 4 °C. The supernatant was then collected and mixed with SDS–PAGE loading buffer under reducing conditions. Separated proteins were transferred to nitrocellulose membranes and probed with appropriate antibodies in 5% non-fat milk powder in PBS-0.05% Tween20. Bound secondary peroxidase-conjugated antibodies were visualized using the ECL system.

**Plaque assay.** Confluent HFF cells were infected with freshly egressed parasites and treated ± ATc if necessary. After 7–10 days, the cells were fixed with PFA/GA and stained with a crystal violet solution (Sigma).

**Intracellular growth and synchronicity of division assays.** Freshly egressed parasites were inoculated on confluent HFFs and allowed to grow for 24 h before fixation with PFA/GA. For MycUNC-iKD parasites, a pre-treatment ± ATc was performed 24 h before egress and continued for 24 h before fixation with PFA/GA.

For intracellular growth assay, IFAs were performed using α-GAP45 and the number of parasites per vacuole was determined by counting the parasites in 100 vacuoles in duplicate for three independent experiments. The data are presented as mean ± s.d.

To determine the synchronicity of cell division within the vacuoles, double-labelling IFAs were performed using α-ISP1 and α-IMC1 antibodies. The development of the daughter cells was evaluated in 100 vacuoles in duplicate for three independent experiments. For type I RHΔKU80, MyoI-KO and MyoJ-KO, fixation has been done 24 and 30 h post invasion, for type II strains, fixation has been done 30 and 42 h post invasion. The results are presented as mean ± s.d.

**Competition assay.** Type I and type II WT, MyoI-KO and MyoJ-KO parasites were mixed with GFP-expressing parasites at a ratio of about 80/20. This ratio was then determined over 10 passages by IFA using α-GAP45. At each passage, 100 vacuoles were counted in duplicate from three biological replicates. The ratios have been normalized to 80% at $t_0$. The data are presented as mean ± s.d.

**Red/green invasion assay.** MycUNC-iKD parasites were pre-treated for 48 h ± ATc before performing the assay as previously described[9]. The number of intracellular and extracellular parasites was determined by counting 100 parasites in duplicate for three independent experiments. The results are presented as mean ± s.d.

**Induced egress assay.** Freshly egressed MycUNC-iKD parasites were inoculated on HFF cells and allowed to grow for 30 h ± ATc before adding either the calcium ionophore A23187 (3 μM) or DMSO for 7 min as previously described[9]. Double-labelling IFA was performed using α-GRA3 and α-GAP45 antibodies. The average number of egressed vacuoles was determined by counting 100 vacuoles in duplicate for each condition and for three independent experiments. The results are presented as mean ± s.d.

**Induced gliding assay.** MycUNC-iKD parasites were grown for 48 h ± ATc. Freshly egressed parasites were settled on poly-L-lysine-coated coverslips in DMEM by centrifugation 1 min at 1,000 r.p.m. and then incubated for 15 min in an HEPES/calcium-saline solution with calcium ionophore A23187 (3 μM) before fixation with PFA/GA. α-SAG1 antibody was used without permeabilization to visualize the trails and the parasites. Three independent experiments have been performed.

**Apicoplast segregation assay.** Freshly egressed RHΔKU80 and MycUNC-iKD parasites, pre-treated ± ATc for 24 h, have been inoculated on new HFFs and allowed to grow for 24 h ± ATc before fixation with PFA/GA. Double-labelling IFA was performed using α-Cpn60 and α-actin antibodies. The number of vacuoles with a correct segregation of the apicoplast was determined by counting the parasites and the apicoplasts in 100 vacuoles in duplicate for three biological replicates. The data are presented as mean ± s.d.

**Delayed death phenotype.** Freshly egressed parasites of the DD-MyoF-tail, DD-MyoF-tail/MyoI-KO and DD-MyoF-tail/MyoJ-KO strains were mixed with GFP-expressing parasites at a ratio of 80/20. These extracellular parasites were then treated for 3 h ± Shield-1 before allowing them to invade new HFFs under the same treatment. The ratio of non-GFP/GFP parasites was then assessed intracellularly at 24 and 72 h and extracellularly at 48 h. Hundred vacuoles or parasites in duplicate from four independent experiments were counted. The ratios non-GFP/GFP have been normalized to 80/20 at 24 h. The results are presented as mean ± s.d.

**TEM.** Freshly egressed parasites (KO-MyoI and KO-MyoJ) were inoculated on confluent HFFs and allowed to grow for 24 h while UNC-iKD parasites were first pre-treated ± ATc for 24 h before egress and then inoculated on new confluent HFFs and allowed to grow for 24 h ± ATc. Infected host cells were washed with 0.1 M PB pH 7.4, then fixed with 2.5% glutaraldehyde in 0.1 M PB pH 7.4, scrapped and pelleted. Samples were then treated as previously described[10]. Thin sections were analysed using a Technai 20 electron microscope (FEI Company) at the 'Pôle

Facultaire de Microscopie Ultrastructurale (PFMU)' of the Faculty of Medicine of Geneva.

**Serial sections TEM.** HFF cells infected with RHΔKU80 parasites were grown in HFF monolayer on round (12 mm) glass coverslips for 24 h. Cells were fixed with 2.5% GA/2% PFA (Electron Microscopy Sciences) in 0.1 M phosphate buffer (PB) at pH 7.4 for 1 h at room temperature. Cells were extensively washed with 0.1 M cacodylate buffer, pH 7.4 and post fixed with 1% osmium tetroxide (Electron Microscopy Sciences) and 1.5% potassium ferrocyanide in 0.1 M cacodylate buffer, pH 7.4 for 40 min followed by 1% osmium tetroxide (Electron Microscopy Sciences) alone in 0.1 M cacodylate buffer pH 7.4 for additional 40 min. Cells were then washed twice for 5 min in double distilled water and *en block* stained with aqueous 1% uranyl acetate (Electron Microscopy Sciences) for 1 h. After 5 min wash in double-distilled water, cells were dehydrated in graded ethanol series (2× 50%, 70%, 90%, 95% and 2× absolute ethanol) for 3 min each wash. Cells were then infiltrated with graded series of Durcupan resin (Electron Microscopy Sciences) diluted with ethanol at 1:2, 1:1 and 2:1 for 30 min each, and twice with pure Durcupan for 30 min each. Cells were infiltrated with fresh Durcupan resin for additional 2 h and the coverslips were placed with grown cells faced down on 1 mm-thick silicone ring used as spacer placed on a glass slide coated with mould-separating agent and filled with fresh resin. This sandwich was polymerized at 65 °C overnight to cure the resin. The glass coverslips were removed from the cured resin by immersing the polymerized resin disk alternately into hot (60 °C) water and liquid nitrogen.

On the exposed surface of the resin block, the position of selected cells with PVMs of parasites was marked using a laser microdissection microscope (Leica Microsystems) to help cut out the cells from resin disk and to facilitate the trimming process. The selected area was cut from the disk using a single edged razor blade and was glued with superglue to a blank resin block. The cutting face was trimmed using a Leica Ultracut UCT microtome (Leica Microsystems) and a glass knife. Ultrathin serial sections (70 nm) were cut with a diamond knife (DiATOME) and collected onto 2 mm single slot copper grids (Electron Microscopy Sciences) coated with Formvar support film.

Sections were examined using a Tecnai 20 TEM (FEI) operating at an acceleration voltage of 80 kV and equipped with a side-mounted MegaView III CCD camera (Olympus Soft-Imaging Systems) controlled by iTEM acquisition software (Olympus Soft-Imaging Systems).

The diameter of the connection was measured on several sections of three vacuoles containing 8, 8 and 16 parasites. A total of 111 measurements have been done with $n=24$ and 37 for the 2 first vacuoles and $n=50$ for the vacuole containing 16 parasites. These measurements provided a diameter of $322 \pm 53$ nm (mean ± s.d.) for the connectivity between the parasites.

**3D reconstruction of TEM images.** Serial images through the selected PVs were combined into single image stacks and aligned using the FIJI programme (fiji.sc/). Segmentation of the feature of interest and 3D reconstructions were done using the TrakEM2 plugin for FIJI, and final 3D models were visualized using the Blender programme (v.2.79; www.blender.org/).

**Isolation and differentiation of BMDMs.** BMDMs were obtained from two 6-weeks-old CD1 female mice (Charles River laboratories) by flushing marrow from the hind tibias and femurs using a 25G needle. The cell suspension was passed through an 18G needle to disperse cell clamps. Cells were cultured in DMEM medium supplemented with 10% FCS, 100 U ml$^{-1}$ penicillin, 0.1 mg ml$^{-1}$ streptomycin and 20% L929-cell supernatant containing M-CSF at 37 °C with 5% CO$_2$ in humidified air. Non-adherent cells were passed the next day to 10-cm bacteriological Petri dishes (2–3 × 10$^6$ cells per dish) and collected for experiments 6 days later using a cell scraper in ice-cold PBS and replated for 4 h before infection. BMDMs were infected (MOI = 1) for 24 h with RH-GFP or 48 h with ME49-GFP before imaging.

**Harvest of peritoneal contents and imaging of *in vivo* infected cells.** A 6-weeks-old CD1 female mouse (Charles River laboratories) was i.p. infected with 10$^6$ parasites and killed on day 3 p.i., immediately followed by a peritoneal lavage with 5 ml of PBS. A measure of 4 ml of recovered lavage fluid were centrifuged at 250g for 10 min and the cell pellet was resuspended in 100 μl of PBS. A fraction of 10 μl was settled on a glass slide under a coverslip with the edges sealed for FRAP imaging.

**Virulence in mice.** CBA/j mice (female, 6 weeks, Charles River Laboratories) were infected by intraperitoneal injection. The health of the mice was monitored daily until they presented severe symptoms of acute toxoplasmosis (bristled hair and complete prostration with incapacity to drink or eat) and were killed on that day.

**Processing of the brains and tissue cysts counting.** Each brain was homogenized in 1 ml PBS with 1% Tween and syringe passaged five times through a 16G needle to break up large clumps. Then, the homogenate was sequentially

syringe passaged through an 18G needle (5 times), a 20G needle (10 times) and a 23G needle (10 times). Tissue cysts number was estimated by counting five fractions of 10 μl from each brain homogenate using the ×20 objective of an inverted microscope.

**Ethics statement.** All animal experiments were conducted with the authorization number 1026/3604/2, GE30/l3 according to the guidelines and regulations issues by the Swiss Federal Veterinary Office. No human samples were used in these experiments.

**Data availability.** All relevant data are available from the authors on request.

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

## Acknowledgements

We gratefully acknowledge the assistance and support of Jean-Baptiste Marq for the documentation of EM pictures and Julien Salamun for his early contribution to MyoJ characterization (CMU, Faculty of Medicine, University of Geneva). Dr Maryse Lebrun (University of Montpellier) is acknowledged for providing the LIC-Centrin2-YFP-CAT vector, Dr Markus Meissner (University of Glasgow) for providing the U1 vector, Dr David Sibley (Washington University) for the CRISPR/Cas9 plasmid, Dr Mohamed-Ali Hakimi for the SET8 vector (University of Grenoble) and Yalin Emre for reagents (M-CSF).

This work was supported by the Swiss National Foundation to D.S.-F. (FN3100A0-116722 and CRSII3_160702). D.S.-F. is an International Scholar of the Howard Hughes Medical Institute. K.F. received funding from the 'Sir Jules Thorn Charitable Overseas Trust reg., Schaan' subsidy for young researchers. Results incorporated in study received funding from the European Research Council (ERC) under the European Union's Horizon 2020 research and innovation programme under Grant agreement no. 695596.

## Author contributions

K.F., D.J. and D.S.-F. designed experiments; K.F., D.J., A.G., P.-M.H. and B.M. performed experiments; K.F., D.J., A.G., P.-M.H. and D.S.-F. analysed the data; K.F., D.J., D.S.-F. wrote the manuscript.

## Additional information

**Competing interests:** The authors declare no competing financial interests.

