## [Peer Review File · Nature Communications]

Reviewers' comments:

Reviewer #1 (Remarks to the Author):

Summary

In this manuscript the authors undertake a systematical investigation of apicomplexan myosins in the parasite *Toxoplasma gondii*. *Toxoplasma* expresses 11 myosin heavy chains and hereby poses the largest repertoire among the Apicomplexa with seven out of the 11 genes being uncharacterized as yet. Germane to the function of myosins, the authors first functionally dissect a previously described myosin folding co-chaperone, TgUNC. TgUNC is essential for parasite survival and its conditional down regulation reveals it to be a prerequisite for most all myosin heavy chains during the lytic cycle. It further exhibits several phenotypes previously described for individual myosins and, importantly, uncovers two novel described myosin functions, namely involvement of posterior pole constriction and parasite organization in the vacuole.

The authors identify the unconventional myosin, MyoJ, to be responsible for the larger width of the posterior pole and described the unconventional myosin MyoI together with MyoJ to be involved in parasite cell-cell communication inside the vacuole as well as formation of the residual body. The authors' observation on connecting the loss of synchronous cell division within a vacuole with cell-cell communication (observed through synchronous cell division cycles) and even the delayed death phenotype observations are laudable and impressive as these and similar phenotypes have been around for a while yet the connection with cell-cell communication has never been made. By also adding mechanism through myosin contribution, this paper is a major leap in our understanding of the *Toxoplasma* lytic cycle. Furthermore, while MyoJ is non-essential in the lytic cycle of the parasite, it is found to be critical for latent infection of the cyst-forming *Toxoplasma* strain ME49 highlighting a crucial function of either posterior pole contraction or cell-cell communication once the tachyzoites switches to the dormant bradyzoite stage.

Overall this manuscript is well written and the data are convincing and plausible presented. I only have relatively minor comments.

Specific comments:

1. Line 140, Fig S2A. The GRA3 signal looks the same in both DMSO and A23187 treated samples. This image does not permit the observation that "the PV membrane (PVM) was clearly ruptured.
2. Can a role for Cen2 technically be excluded in basal complex constriction? Specially when compared to the presented images in Fig 2 I (UNC cKO and Centrin2) and 4 A (MyoJ KO and Centrin2) a difference in cell width is. Does this represent an additional defect observed after myoJ KO? On the other hand, inclusion of IFAs using the marker of mature parasites GAP45 or the plasma membrane marker SAG1 could help to resolve this question and highlight only mature parasites. See also #3
3. while the used marker IMC1 is present in the mother as well as in the daughter parasite, staining of the mature cell as presented in Fig 2 K (+ATc) for example is weak. Considering this, do the parasites presented in Fig 2 G and H (UNC cKO and MyoC, UNC cKO and MORN1) and Fig 4 A (MyoJKO and MyoC) represent fully mature parasites or daughter parasites at the end of division? A measured quantification of basal width would overcome this concern.
4. Fig S3. These are not complete KOs. Only expressing the head domain could result in dominant negative effects. Although this is very briefly mentioned, a bit more of a contingency might be in place. However, the phenotypes do recapitulate the UNC KO phenotype quite well so not the biggest concern.
5. MyoJ has recently been described to be only weakly related to myosins of class VI and was therefore suggested for an Apicomplexa-specific category, myosin XXIII (Sebe-Pedros et al 2014, Genome Biol Evol, doi: 10.1093/gbe/evu013). The authors should include this information, which potentially underlines a poor structural similarity of this specific myosin and might indicate a possible divergent mode of function unique to Apicomplexa.

Reviewer #2 (Remarks to the Author):

The manuscript entitled "Myosin-dependent cell-cell communication controls synchronicity of division in acute and chronic stages of *Toxoplasma gondii*" is largely the characterization of unknown myosins and associated factors (Fig. 1-3; Fig. S1-4). These experiments are well done and they fill a knowledge gap for *Toxoplasma*. What started as a basic characterization of myosin function that confirmed what was already known for some myosins led to an interesting discovery of a potential mechanism for intravacuolar cell-cell communication. Genetic approaches established that MyoI and MyoJ may have key roles in this putative communication mechanism, which is proposed to explain two remarkable features of tachyzoite replication biology; synchronous intravacuolar growth and the formation of rosettes.

That basic studies of myosins would unexpectedly discover such a novel mechanism is what we all wish for our labs. However, the possible rush to reach a conclusion here has overlooked some important questions.

1) It is puzzling that the authors have tachyzoite cultures that are 100% synchronous. Even using synchronization methods 100% synchrony is not really achievable and synchrony in these methods breaks down after a single division. In normal tachyzoite cultures intravacuolar synchrony is highest in the first two divisions and then breaks down in late vacuoles. Does the transport of GFP between tachyzoites also follow this growth progression?

2) The authors should further address the remarkable restoration of GFP fluorescent after FRAP. GFP has to pass at least 2 bilayers and diffuse twice in thick cytoplasm and once in nucleoplasm in order to achieve a distribution equal between all 4 cells in just one minute. How does GFP know that it has to go to that specific "empty" cell? How does the parasite know that equilibrium of a fluorescent protein is reached? Non-fluorescent GFP is still there. The authors should discuss their proposed mechanism in terms of known rates of protein transport in eukaryotes. Does this rapid transport mechanism occur with other cytoplasmic proteins?

3) MyoJ-knockout mutant has an open basal end, whereas, wild type parasites have a restricted basal end. The proposed model for cell-cell communication requires a transport path through the constricted basal end of one parasite then through the RB matrix and back through the basal end restriction of the second parasite. Yet, the open basal end of the MyoJ-knockout blocks this transport path. It is not clear how this result squares with the model. The authors should comment on what mechanism would effectively use the basal end as a "stargate" to achieve rapid protein transport from one parasite to another.

4) The importance of the cell-cell communication model to native *Toxoplasma* biology is a concern given that disruption of the mechanism seems to have no real consequence on virulence, growth rates, or bradyzoite differentiation. The growth defects of the MyoJ KO appear to be independent of the cell-cell communication mechanism, which is disrupted in the MyoI KO strain that shows no growth defects. It is important to demonstrate that cell-cell communication is not an oddity of infections of human fibroblasts forced to grow as a single monolayer on plastic. Does cell-cell communication occur if the tachyzoites are grown in mouse macrophages? Is there evidence for cell-cell communication in infected host cells obtained from the peritoneum of mice. Obtaining these infected cells from mice is not difficult, especially for the RH strains.

Reviewer #3 (Remarks to the Author):

The paper by Frenal et al reports on the function of two myosins in the parasite *Toxoplasma gondii*. Through gene deletions and several assays the authors show a function of these myosins in closure of the daughter parasites during intracellular growth. While their data supports these findings, the authors over-interpret their findings as revealing a role for these myosins in cell-cell communication. This is a pity as this paper includes a lot of work. It is clearly shown that proteins can diffuse between cells in wild type parasites but not in parasites lacking the two myosins. But this in no way ascertains a role for cell-cell communication in playing a role for synchronicity of division. Further the impact of the lost synchronicity is weak and can only be found in one strain of parasites in vivo, probably the more important one, as it causes chronic toxoplasmosis, while the strain that shows no phenotype causes only acute disease. Yet, the authors show that the chronic stages do multiply asynchronously. Hence this reviewer misses the connection between the data and the conclusion. The phenotype might well be one of reduced fitness in the many processes playing a role during in vivo infection. This should be critically discussed rather than over-interpreting the results as revealing some form of communication. This reviewer would have preferred some molecular functional insights into how the myosins exert their specific function, which appears to be actin-independent and thus remains somewhat mysterious.

Major comments:

The authors make many different parasites strains, why not one lacking both myosins, MyoI and MyoJ? Maybe the phenotype would be much stronger.

The plaque assays in Figure 3B and 5D appear to show different results for MyoJ-KO, once there is an effect, once not. How can that be? Do the parasites in 5B grow better because they also lack a kinase?

The title and abstract should not include cell-cell communication. The paper reveals connectivity, not communication. Similarly it should be removed from the rest of the text. It might, of course, be added to a speculative section in the discussion.

Lines 444-452: Mechanical resistance is discussed out of the blue. Yet, this might make much more sense than cell-cell communication as the real cause of the phenotype. I don't understand how the competition assay gives an insight into mechanical resistance. And how can an in vitro assay provide a difference, when an in vivo assay cannot? The authors should try an informative assay to investigate mechanical resistance, maybe as simple as infections in low or high salt conditions/changed osmolarity or by using cells or cell systems (e.g. trans-well assays) that provide a stronger barrier for invasion/egress. If positive this might give the paper the functional insight this reviewer is currently missing. Alternatively the authors (following the speculation on RB sealing during egress, line 463) might film the parasites during egress and see differences in viability of parasites.

Figure 7B is suggesting an intriguing difference between the two situations on the right. Could the different distribution of metabolites be visualized using an antibody and FACS or simply by immunofluorescence assay? Or a similar assay? This figure would be easier to read if the title would simply say, which proteins are there and which are missing.

Minor comments/suggestions:

Please tone down conclusions in line 256

In line 292 no causal connection between diffusion and division was established in the paper. This needs to be made very clear.

295: how can CytD not depolymerize actin filaments?

425: The FRAP experiments quite beautifully show when parasites are connected to each other but they do not show in any way anything about cyclins.

430: Plasmodium appears to divide in a completely different way, that makes this statement superfluous.

Figure 1 could go to the supplement and could be replaced by a table showing the percentages of

knock-downs at 24 and 48 hours in the main text.

Figure 2 G, H, J and Figure 4A, C would benefit from quantification of the reported effects.

Figure 3 would be easier to read if the same color would be used for the anti-Ty staining.

Typos/writing:

47: delete 'importantly'

101: replace 'proved' with 'shown' or similar

137: in 'a' gliding assay

148: the microtubules making up the conoid appear very distorted and I am not sure they should be called microtubules

170: or 'a' cytokinesis defect

288: only 'a' subset

290: please show the data

388: MyoE

454: Tg MyoJ-mediated basal complex constriction and TgMyoI-mediated parasite connectivity.

Reviewer #1

Summary

In this manuscript, the authors undertake a systematical investigation of apicomplexan myosins in the parasite *Toxoplasma gondii*. *Toxoplasma* expresses 11 myosin heavy chains and hereby possesses the largest repertoire among the Apicomplexa with seven out of the 11 genes being uncharacterized as yet. Germane to the function of myosins, the authors first functionally dissect a previously described myosin folding co-chaperone, TgUNC. TgUNC is essential for parasite survival and its conditional down regulation reveals it to be a prerequisite for most all myosin heavy chains during the lytic cycle. It further exhibits several phenotypes previously described for individual myosins and, importantly, uncovers two novel described myosin functions, namely involvement of posterior pole constriction and parasite organization in the vacuole.

The authors identify the unconventional myosin, MyoJ, to be responsible for the larger width of the posterior pole and described the unconventional myosin MyoI together with MyoJ to be involved in parasite cell-cell communication inside the vacuole as well as formation of the residual body. The authors' observation on connecting the loss of synchronous cell division within a vacuole with cell-cell communication (observed through synchronous cell division cycles) and even the delayed death phenotype observations are laudable and impressive as these and similar phenotypes have been around for a while yet the connection with cell-cell communication has never been made. By also adding mechanism through myosin contribution, this paper is a major leap in our understanding of the *Toxoplasma* lytic cycle. Furthermore, while MyoJ is non-essential in the lytic cycle of the parasite, it is found to be critical for latent infection of the cyst-forming *Toxoplasma* strain ME49 highlighting a crucial function of either posterior pole contraction or cell-cell communication once the tachyzoites switches to the dormant bradyzoite stage.

Overall this manuscript is well written and the data are convincing and plausible presented. I only have relatively minor comments.

Specific comments:

1. Line 140, Fig S2A. The GRA3 signal looks the same in both DMSO and A23187 treated samples. This image does not permit the observation that "the PV membrane (PVM) was clearly ruptured".

We agree that the PVM was not clearly visible on the merge picture. We therefore updated the new **Figure S2A** by presenting GRA3 signal in an independent panel for more clarity. In addition, we provide a second example of vacuoles treated either with the calcium ionophore A23187 or with DMSO.

2. Can a role for Cen2 technically be excluded in basal complex constriction? Specially when compared to the presented images in Fig 2I (UNC-iKD and Centrin2) and 4A (MyoJ-KO and Centrin2) a difference in cell width is. Does this represent an additional defect observed after MyoJ-KO? On the other hand, inclusion of IFAs using the marker of mature parasites GAP45 or the plasma membrane marker SAG1 could help to resolve this question and highlight only mature parasites. See also #3

Indeed, a direct role for centrin2 (CEN2) in the basal complex constriction could not be excluded, quite the opposite. To resolve this question, we have generated an inducible parasite line **CEN2-iKD** and shown that after 48 hr of ATc treatment, parasites depleted in CEN2 exhibit the same phenotype as MyoJ-KO, e.g. a defect in the constriction of their basal pole without defect in cytokinesis. In addition and importantly, we demonstrated that this constriction is truly an actomyosin-dependent process as parasites treated with cytochalasin D displayed a larger basal cup (**Figure 4G**). Furthermore, a conditional knockout of actin revealed the same phenotype (Periz J et al, eLife in press, M. Meissner pers. communication). The functional relationship between CEN2 and MyoJ is not yet fully understood. MyoJ apparently positions/targets CEN2 to the basal cup and at the same time CEN2 could be critical for the function of the motor. This is conceivable since CEN2 is small EF hand containing protein, which could potentially act as a myosin light chain for MyoJ and modulate the motor function. The new data on CEN2 function have been added to the new **Figure 4 (panels D-F)** and **Figure S3D** and are discussed in the text. See also #3 for the second part of the answer.

3. While the used marker IMC1 is present in the mother as well as in the daughter parasite, staining of the mature cell as presented in Fig 2K (+ATc) for example is weak. Considering this, do the parasites presented in Fig 2G and H (UNC-iKD and MyoC, UNC-iKD and MORN1) and Fig 4A (MyoJ-KO and MyoC) represent fully mature parasites or daughter parasites at the end of division? A measured quantification of basal width would overcome this concern.

We can be sure that the parasites on these panels are mature parasites as only one ring of MyoC is observed (Fréna *et al.*, 2014). In addition and as suggested to overcome this concern, we measured the diameter of the basal ring using the signal of endogenously tagged MyoC in fully mature parasites only. We determined that this diameter is about $0.93 \pm 0.10 \mu\text{m}$ in untreated MycUNC-iKD (n=25), $1.66 \pm 0.16 \mu\text{m}$ in MycUNC-iKD + ATc (n=37) and $1.70 \pm 0.16 \mu\text{m}$ in MyoJ-KO parasites (n=37). These defects are comparable and can be directly attributed to the destabilization of MyoJ in MycUNC-iKD. The same methodology was applied on CEN2-YFP-iKD. The observed defects were similar ($0.88 \pm 0.7 \mu\text{m}$ and $1.48 \pm 0.13 \mu\text{m}$ in untreated and treated parasites, respectively) and reflect again that MyoJ and CEN2 are both involved in the constriction of the basal complex. These data have been incorporated in the new **Figure S2F**.

4. Fig S3. These are not complete KOs. Only expressing the head domain could result in dominant negative effects. Although this is very briefly mentioned, a bit more of a contingency might be in place. However, the phenotypes do recapitulate the UNC KO phenotype quite well so not the biggest concern.

As mentioned in the result part, the KOs reported in Fig. S3 and Fig. 3 are truncated versions of the myosins. We purposely interrupted the gene in the part coding for the head domain in order to not express IQ domains that could indeed create a dominant negative effect by titrating myosin light chain(s) of other myosins. As indicated in the scheme Fig S3A, we introduced in frame 3xTy-tags at the end of the head domains and none of these truncations have been detected either by IFA or western blot, suggesting that the truncated proteins are unstable.

In regard to MyoI-KO and MyoJ-KO, which exhibit noticeable phenotypes, the corresponding genes have been subsequently disrupted using a CRISPR/Cas9 strategy with gRNAs positioned closer to the ATG where the DHFR resistance cassette was inserted. These independent KOs show the same phenotypes as the corresponding truncated version reported in Fig 3.

5. MyoJ has recently been described to be only weakly related to myosins of class VI and was therefore suggested for an Apicomplexa-specific category, myosin XXIII (Sebe-Pedros *et al* 2014, Genome Biol Evol, doi: 10.1093/gbe/evu013). The authors should include this information, which potentially underlines a poor structural similarity of this specific myosin and might indicate a possible divergent mode of function unique to Apicomplexa.

In the article of Sebe-Pedros *et al.*, the authors used a paneukaryotic whole-genome approach including some Alveolata myosins sequences, although the TgMyoI class XXIV and TgMyoK class XXIII have not been included, and propose that MyoJ belongs to the class XXIII. This Alveolata specific class XXIII is suggested to be part of the myosin VI-like class that encompasses the class VI, XXIII, XXX and XXXII (Fig. S2 of Sebe-Pedros *et al.*) demonstrating a close proximity between these classes and underlining the difficulty to solve the phylogenetic relationships depending of the dataset and the phylogenetic approaches. We thank the reviewer for pointing this out and have included this study in the references and modified the text accordingly.

Reviewer #2

The manuscript entitled "Myosin-dependent cell-cell communication controls synchronicity of division in acute and chronic stages of *Toxoplasma gondii*" is largely the characterization of unknown myosins and associated factors (Fig. 1-3; Fig. S1-4). These experiments are well done and they fill a knowledge gap for *Toxoplasma*. What started as a basic characterization of myosin function that confirmed what was already known for some myosins led to an interesting discovery of a potential mechanism for intravacuolar cell-cell communication. Genetic approaches established that MyoI and MyoJ may have key roles in this putative communication mechanism, which is proposed to explain two remarkable features of tachyzoite replication biology; synchronous intravacuolar growth and the formation of rosettes.

That basic studies of myosins would unexpectedly discover such a novel mechanism is what we all wish for our labs. However, the possible rush to reach a conclusion here has overlooked some important questions.

1) It is puzzling that the authors have tachyzoite cultures that are 100% synchronous. Even using synchronization methods 100% synchrony is not really achievable and synchrony in these methods breaks down after a single division. In normal tachyzoite cultures intravacuolar synchrony is highest in the first two divisions and then breaks down in late vacuoles. Does the transport of GFP between tachyzoites also follow this growth progression?

It is correct that tachyzoite cultures are never synchronous but we are not referring here to synchronicity within the parasite population but exclusively to synchronized parasites within a given vacuole (intravacuolar parasites). In that case intravacuolar parasites were previously reported to be synchronized. "*The parasites in a small vacuole can be accurately counted and they divide in synchrony for at least the first four cycles after invasion*" (Hu et al., 2004).

To make this point clearer we have included in the new **Figure S5A** supplementary data showing that most of large vacuoles (16 and 32 parasites) are synchronized (using the daughter cell marker ISP3) until the rosette organization is compromised. In addition, we show by FRAP that GFP is transported between parasites in these large vacuoles. Interestingly, in some big vacuoles, few parasites lost the connection while the rest of the rosette is still connected, movies S2.

2) The authors should further address the remarkable restoration of GFP fluorescent after FRAP. GFP has to pass at least 2 bilayers and diffuse twice in thick cytoplasm and once in nucleoplasm in order to achieve a distribution equal between all 4 cells in just one minute. How does GFP know that it has to go to that specific "empty" cell? How does the parasite know that equilibrium of a fluorescent protein is reached? Non-fluorescent GFP is still there. The authors should discuss their proposed mechanism in terms of known rates of protein transport in eukaryotes. Does this rapid transport mechanism occur with other cytoplasmic proteins?

Reading these comments, we feared that we failed to explain with sufficient clarity the nature of the connection that exists between intravacuolar parasites. GFP does not need to cross any lipid bilayer since parasites are sharing the same cytosol at the posterior pole via connected tubes. In consequence, the restoration of fluorescence is simply due to the **diffusion** of GFP. To be more explicit in regard to the nature of the connection between intravacuolar parasites, we have performed 3D reconstitution of a vacuole containing wild type parasites after imaging serial sections by transmission electron microscopy in order to visualize the structure that cannot be seen by IFA. The data are now presented in the new **Figure 7** as well as in the **movies S6 and S7**.

Most interestingly, the sections revealed that the mitochondrion is often present in the connection (RB). In consequence we tested mitochondrial protein diffusion by FRAP using a protein targeted to the matrix of the mitochondrion (N-term-SOD2-GFP, Pino *et al.*, 2007). We recorded during 15 minutes but failed to observe a recovery of fluorescence. These data have been discussed in the text and are presented as movie S8. While diffusion between parasites of proteins within the mitochondrial matrix was not expected, in contrast the transmission of electric or electrochemical signals could occur via the mitochondrion found in the connection. This represents an exciting topic, which might deserve further investigations outside the scope of this study that has been mentioned in the discussion.

Importantly we now present here the diffusion SET8, a protein accumulating into the nucleus and expressed in a cell cycle-dependent fashion. Since the diffusion depends on the shuffling of the protein between the cytosol and nucleus, the kinetics of fluorescence recovery is expectedly slower than for GFP but faster than **neosynthesis**, Fig 6H and movie S4.

3) MyoJ-knockout mutant has an open basal end, whereas, wild type parasites have a restricted basal end. The proposed model for cell-cell communication requires a transport path through the constricted basal end of one parasite then through the RB matrix and back through the basal end restriction of the second parasite. Yet, the open basal end of the MyoJ-knockout blocks this transport path. It is not clear how this result squares with the model. The authors should comment on what mechanism would effectively use the basal end as a "stargate" to achieve rapid protein transport from one parasite to another.

MyoJ-KO parasites cumulate 2 defects: a defect in basal constriction (without a defect in cytokinesis) due to the loss of MyoJ/CEN2 and a mislocalization of MyoI. The defect in intravacuolar connection is most likely due to this mislocalization of MyoI which we demonstrated to be essential to maintain this connectivity.

4) The importance of the cell-cell communication model to native *Toxoplasma* biology is a concern given that disruption of the mechanism seems to have no real consequence on virulence, growth rates, or bradyzoite differentiation. The growth defects of the MyoJ KO appear to be independent of the cell-cell communication mechanism, which is disrupted in the MyoI KO strain that shows no growth defects. It is important to demonstrate that cell-cell communication is not an oddity of infections of human fibroblasts forced to grow as a single monolayer on plastic. Does cell-cell communication occur if the tachyzoites are grown in mouse macrophages? Is there evidence for cell-cell communication in infected host cells obtained from the peritoneum of mice. Obtaining these infected cells from mice is not difficult, especially for the RH strains.

It is absolutely correct that the loss of virulence observed with MyoJ-KO parasites does not relate to loss of connectivity since MyoI-KO parasites are unaltered in virulence. The importance of cell-cell communication for *T. gondii* tachyzoites is an intriguing issue. Importantly, the fast expanding asexual stages of Apicomplexa divide by a mechanism referred to merogony/schizogony where the cell increases in size while repeatedly replicating its nucleus and other organelles. *Toxoplasma* (also *Cryptosporidium*, which possesses a MyoI) appears to have developed a strategy to maintain cell-cell communication as a way to synchronize and thus optimize the rapid expansion of fast dividing tachyzoites. This type of connection contrasts with schizogony and offers the unique option to be interrupted to favor slow and asynchronized replication, as requested for the slow growing bradyzoites aiming at encystation.

To demonstrate that cell-cell communication is not an oddity of infections of human fibroblasts, we have infected a mouse with an RH-GFP strain and collected cells from the peritoneum 4 days post infection. Again we observed connectivity. These data are now presented in the new **Figure 6I** and **movie S5**.

Furthermore, we have performed FRAP experiments using differentiated mouse bone marrow derived macrophages (BMDM). In BMDMs infected with a type I RH-GFP strain, parasites were connected. However and interestingly, in BMDMs infected with a type II ME49-GFP, most of the parasites had lost the connection. Strikingly, the same parasites were mostly connected in BMDMs that were switched to non-containing M-CSF medium 24h prior infection. This difference in connectivity is likely due to the known differences in host-parasite interactions between type I and type II strains. These data are now discussed and presented in the new **Figure 9**.

Reviewer #3

The paper by Frenal et al reports on the function of two myosins in the parasite *Toxoplasma gondii*. Through gene deletions and several assays the authors show a function of these myosins in closure of the daughter parasites during intracellular growth. While their data supports these findings, the authors over-interpret their findings as revealing a role for these myosins in cell-cell communication. This is a pity as this paper includes a lot of work. It is clearly shown that proteins can diffuse between cells in wild type parasites but not in parasites lacking the two myosins. But this in no way ascertains a role for cell-cell communication in playing a role for synchronicity of division. Further the impact of the lost synchronicity is weak and can only be found in one strain of parasites in vivo, probably the more important one, as it causes chronic toxoplasmosis, while the strain that shows no phenotype causes only acute disease. Yet, the authors show that the chronic stages do multiply asynchronously. Hence this reviewer misses the connection between the data and the conclusion. The phenotype might well be one of reduced fitness in the many processes playing a role during in vivo infection. This should be critically discussed rather than over-interpreting the results as revealing some form of communication.

This reviewer would have preferred some molecular functional insights into how the myosins exert their specific function, which appears to be actin-independent and thus remains somewhat mysterious. The fact that MyoI- and MyoJ- dependent processes are poorly inhibited by cytochalasin D (CytD) does not imply that the function of these myosins is actin-independent. While CytD prevents actin polymerization, it is not an actin-depolymerizing agent. In consequence events that depend on stable actin filaments would be less efficiently affected by this drug.

That being said, we are grateful to the reviewer for having raised this issue. We have analyzed the role of CytD in more depth on the basal complex constriction. In previous investigations by us, using 200 nM of CytD to assess the role of actin dynamic in apicoplast inheritance (Jacot et al., 2013) and by others (Shaw et al., 2000; Gubbels et al., 2006) the impact on posterior constriction was not observed. Using 1 μ M of CytD and appropriate markers of the basal end such as MyoJ and Cen2 we were able to measure a significant change in constriction upon CytD treatment. This data are reported in the new **Figure 4G**.

The FRAP experiments, on the other hands, showed that only prolonged treatment (8 hours) but not short term treatments (2 hours) with CytD affected this connection (Figure S5C). This discrepancy and the misinterpretation of the role of CytD in the constriction of the basal complex leads us to conclude that filaments associated to TgMyoI and TgMyoJ are either very stable or poorly sensitive/accessible to this actin depolymerizing agent. We have now revised this interpretation. The best plausible explanation here is that at short time points although the actin filaments might be impacted, the membranous connection is not. Therefore, diffusion of soluble protein can still occur. At later time points (8-10 hr treatment), the defect in cell-cell communication becomes visible as parasites divided and could not form/maintain the connection. The implication of actin in these events is corroborated by TgACT-iKO parasites that exhibit an impaired posterior constriction and loss of vacuolar synchronicity as well as the collapsing of the actin network in the residual body upon CytD treatment (Javier Periz et al. in press and available in BiorXiv, 2016).

Major comments:

The authors make many different parasites strains, why not one lacking both myosins, MyoI and MyoJ? Maybe the phenotype would be much stronger.

We generated the MyoI/J-KO and performed a competition assay with WT, MyoI-KO, MyoJ-KO and MyoI/MyoJ-KO. The double MyoI/J-KO did not lead to stronger phenotype. This was anticipated since in absence of MyoJ, MyoI is mis-localized and fails to function in cell-cell communication. These data have been included in the new supplementary **Figure S4D**.

The plaque assays in Figure 3B and 5D appear to show different results for MyoJ-KO, once there is an effect, once not. How can that be? Do the parasites in 5B grow better because they also lack a kinase?

The fitness defect monitored in parasites lacking MyoJ by plaque assay is modest and not suitable for

an accurately quantitative analysis. To overcome this limitation, we performed a competition assay with WT, CDPK2-KO/MyoI-KO and CDPK2-KO/MyoJ-KO that is more sensitive to assess the extent of fitness loss. The analysis indicates that MyoJ-KO does not grow significantly better when CDPK2 gene is deleted as shown in the new **Figure 5C**.

The title and abstract should not include cell-cell communication. The paper reveals connectivity, not communication. Similarly, it should be removed from the rest of the text. It might, of course, be added to a speculative section in the discussion.

We respectfully disagree with the reviewer. When cells are connected by nanotubes it is referred to as a mechanism of cell-cell communication. This connection is responsible for the synchronized division of the parasites and in consequence there are key parasite proteins that diffuse to ensure a concerted control of the cell cycle. To illustrate this point we could not use cyclins that are apparently poorly expressed and not amenable to GFP fusion (M. White, personal communication). Instead, we chose to examine a histone lysine methyltransferase, SET8 which is a nuclear protein reported to be tightly cell cycle regulated and implicated in regulation of gene expression (Sautel et al., 2007). We first show by IFA in MyoI-KO and MyoJ-KO that SET8 is strikingly differentially expressed in parasites within a given vacuole, attesting for loss of synchronicity (new **Figure 6C**). Furthermore, using a GFP-tagged version of SET8, we show by FRAP that in wild type parasites, there is diffusion of this nuclear protein into the neighboring parasites. Expectedly, the time of FRAP recovery was longer given the need of SET8 to shuttle in and out of the nucleus. These data have been including in the study as new **Figure 6H**.

Lines 444-452: Mechanical resistance is discussed out of the blue. Yet, this might make much more sense than cell-cell communication as the real cause of the phenotype. I don't understand how the competition assay gives an insight into mechanical resistance. And how can an in vitro assay provide a difference, when an in vivo assay cannot? The authors should try an informative assay to investigate mechanical resistance, maybe as simple as infections in low or high salt conditions/changed osmolarity or by using cells or cell systems (e.g. trans-well assays) that provide a stronger barrier for invasion/egress. If positive this might give the paper the functional insight this reviewer is currently missing.

We have subjected MyoJ-KO and control parasites to a hypo-osmotic shock and found that the MyoJ-KO parasites were slightly more sensitive to this stress than MyoI-KO and wild type. The results of these experiments are presented in the new **Figure S10**. It is not certain that this fragility explains the dramatic loss in virulence observed with TgMyoJ-KO in ME49 and further investigations will be necessary to address this issue.

Alternatively, the authors (following the speculation on RB sealing during egress, line 463) might film the parasites during egress and see differences in viability of parasites.

In contrast to WT, the MyoI-KO and MyoJ-KO parasites are sealed inside the vacuole as shown by EM analysis. In consequence, we do not expect a loss of viability at the time of egress, which anyway would be difficult to score experimentally.

Figure 7B is suggesting an intriguing difference between the two situations on the right. Could the different distribution of metabolites be visualized using an antibody and FACS or simply by immunofluorescence assay? Or a similar assay? This figure would be easier to read if the title would simply say, which proteins are there and which are missing.

If GFP and SET8 are able to diffuse between parasites, many other proteins can do it as well and so would do a very large set of metabolites. The figure 7B is intending to depict the mechanism by which an unknown metabolite produced by the apicoplast contributes to explain the delayed death phenotype. Although it is likely to be the product of an essential metabolic pathway of the apicoplast (DOXP or FASII) the current study does not pretend to identify it and even less to visualize it using an antibody. In our view this is outside the scope of this work and we have tried to explain better the purpose of this model figure.

Minor comments/suggestions:

Please tone down conclusions in line 256

We have rephrased this sentence and toned it down. We also refer to the new EM data included in the **Figure 5B** where residual bodies can be observed in both MyoI-KO and MyoJ-KO parasites.

In line 292 no causal connection between diffusion and division was established in the paper. This needs to be made very clear.

Using TgSET8 instead of GFP, we proved that key regulators of cell division are diffusing between parasites and thus plausibly explaining why parasites are synchronized when connected.

295: how can CytD not depolymerize actin filaments?

In contrast to Latrunculin, cytochalasin D does not depolymerize F-actin and only prevents actin polymerization (see comparative study: Wakatsuki et al., Journal of Cell Science, 2000).

425: The FRAP experiments quite beautifully show when parasites are connected to each other but they do not show in any way anything about cyclins.

As mentioned above, cyclins turn out not to be proteins of choice to follow by fusion with GFP. We have instead opted for cell cycle regulated, nuclear histone lysine methyltransferase, TgSET8, and shown by FRAP wild type parasites that this protein can diffuse between parasites. The data are presented in the new **Figure 6H**.

430: Plasmodium appears to divide in a completely different way, that makes this statement superfluous.

This issue is discussed when addressing point 4 of Reviewer 2

Figure 1 could go to the supplement and could be replaced by a table showing the percentages of knock-downs at 24 and 48 hours in the main text.

We feel that the raw data presented in figure 1 (clean full western blot analysis) are of high quality, explicit and informative. For example, full length MyoC disappears while a degradation product from the tail is accumulating. We do not see the added value of converting them into processed data.

Figure 2 G, H, J and Figure 4A, C would benefit from quantification of the reported effects.

The measure of the basal width has been performed from IFAs of UNC-iKD +/- ATc and MyoJ-KO using the signal of endogenously tagged MyoC that localizes to the basal ring of mature parasites. We measured the diameter of the basal ring only in fully mature parasites and determined that its diameter is about $0.93 \pm 0.10 \mu\text{m}$ in untreated UNC-iKD (n=25), $1.66 \pm 0.16 \mu\text{m}$ in UNC-iKD + ATc (n=37) and $1.70 \pm 0.16 \mu\text{m}$ in MyoJ-KO (n=37). The same defect attributed to the destabilization of MyoJ was observed in UNC-iKD parasites in presence of ATc. These data have been incorporated in the new **Figure S2F**.

Figure 3 would be easier to read if the same color would be used for the anti-Ty staining.

The suggested change has been made.

Typos/writing:

47: delete 'importantly'

101: replace 'proved' with 'shown' or similar

137: in 'a' gliding assay

148: the microtubules making up the conoid appear very distorted and I am not sure they should be called microtubules

We agree with this comment and we have renamed them **conoid fibers** to be coherent with the publication describing the structure of the conoid (Hu *et al.*, J Cell Biol 2002).

170: or 'a' cytokinesis defect

288: only 'a' subset

290: please show the data

We have now shown the data in **Figure S5C**

388: MyoE

454: Tg MyoJ-mediated basal complex constriction and TgMyoI-mediated parasite connectivity.

All the typos errors/suggestion for changes listed above have been corrected.

REVIEWERS' COMMENTS:

Reviewer #1 (Remarks to the Author):

Summary

The authors have been very responsive to reviewer comment and addressed all major concerns by adding an impressive amount of new experiments and data, making the paper even stronger. Only a few minor comments.

Specific comments:

1. Line 291/5; Fig6C. IFA looks great, but a quantification of the incidence of these phenomena would nail it completely. Although I think the SET-GFP FRAP is even more convincing, but technically a different experiment as in the 6C plasmids are likely transmitted and in 6H it would be protein.
2. Prolonged CytD treatment. Were these experiments done in CytD resistant host cells? I am surprised HFF fibroblasts tolerate such treatment and still result in parasites organized flatly in a vacuole as in the images shown.
3. Rebutal Rev2 #4: "Toxoplasma (also Cryptosporidium, which possesses a MyoI) appears to have developed a strategy to maintain cell-cell communication as a way to synchronize and thus optimize the rapid expansion of fast dividing tachyzoites." Linking the parasite connection to growth rates is an interesting one but does not seem to fit with the MyoI data: MyoI depleted parasites do form normal plaque sizes and in completion do, if anything, outcompete WT parasites (Fig s4d: statistics would be great on this figure for this aspect). So it is unclear how the "optimization" statement fits with the data. I do not believe this statement is made in the paper though.

Reviewer #2 (Remarks to the Author):

The revised manuscript entitled "Myosin-dependent cell-cell communication controls synchronicity of division in acute and chronic stages of *Toxoplasma gondii*" has now addressed the outstanding issues raised by our previous review. The insights into the molecular mechanism that enables tachyzoites to coordinate their cell division will be of interest to the field.

Reviewer #3 (Remarks to the Author):

The authors have nicely addressed all my comments and I am happy now also with the communications part- the additional data including SET8-GFP really helped. Well done and congrats to a beautiful study.

Reviewer #1

Summary

The authors have been very responsive to reviewer comment and addressed all major concerns by adding an impressive amount of new experiments and data, making the paper even stronger. Only a few minor comments.

Specific comments:

1. Line 291/5; Fig6C. IFA looks great, but a quantification of the incidence of these phenomena would nail it completely. Although I think the SET-GFP FRAP is even more convincing, but technically a different experiment as in the 6C plasmids are likely transmitted and in 6H it would be protein.

HA-SET8 has previously been reported to be detectable only in a subset of vacuoles due to the cell cycle regulation of the expression of this gene (Sautel *et al.*, 2007). In wild type parasites, the intravacuolar expression of HA-SET8 is “always” homogenous (in >90% of the vacuoles). When expressed in Myo1-KO or MyoJ-KO, HA-SET8 is only detectable in a subset of the parasites within a vacuole (>50% of the vacuoles show this asynchronicity phenotype). At this point we assume that all the parasites still contain comparable amount of transfected plasmids. Indeed, transient expression of GFP resulted with homogenous expression within a vacuole even in absence of TgMyo1 or TgMyoJ.

The transfer of plasmids via the connection is a parameter to take into consideration. However, we have not been able to demonstrate that plasmid diffusion through the connection plays a significant role since the transiently transfected parasites exhibit homogenous expression of constitutively expressed reporters (pT8-GFP) inside a given vacuole in Myo1-KO or MyoJ-KO, in sharp contrast with HA-SET8, which is cell cycle regulated.

2. Prolonged CytD treatment. Were these experiments done in CytD resistant host cells? I am surprised HFF fibroblasts tolerate such treatment and still result in parasites organized flatly in a vacuole as in the images shown.

No, these experiments have been performed on fully confluent (quiescent) HFF cells that can cope with the prolonged CytD treatment. This tolerance to CytD in quiescent cells contrasts with latrunculin treatment (even short), which is deleterious to the cells that rapidly round up and detach.

3. Rebutal Rev2 #4: “Toxoplasma (also Cryptosporidium, which possesses a Myo1) appears to have developed a strategy to maintain cell-cell communication as a way to synchronize and thus optimize the rapid expansion of fast dividing tachyzoites.” Linking the parasite connection to growth rates is an interesting one but does not seem to fit with the Myo1 data: Myo1 depleted parasites do form normal plaque sizes and in completion do, if anything, outcompete WT parasites (Fig s4d: statistics would be great on this figure for this aspect). So it is unclear how the “optimization” statement fits with the data. I do not believe this statement is made in the paper though.

This statement is speculative and not supported by the absence of phenotype growth defect as noticed by the reviewer. However, that being said, it is possible that in the rich media of the *in vitro* culture condition, the optimization of resources conferred by connectivity is not necessary. This idea is not mentioned in the text.

Reviewer #2

The revised manuscript entitled "Myosin-dependent cell-cell communication controls synchronicity of division in acute and chronic stages of *Toxoplasma gondii*" has now addressed the outstanding issues raised by our previous review. The insights into the molecular mechanism that enables tachyzoites to coordinate their cell division will be of interest to the field.

Reviewer #3

The authors have nicely addressed all my comments and I am happy now also with the communications part- the additional data including SET8-GFP really helped. Well done and congrats to a beautiful study.